# Lipid nanoparticle-encapsulated *Dnai1* mRNA rescues ciliary activity in primary ciliary dyskinesia mouse cell models

Amanda J. Smith[1,2], Patrick R. Sears[1], Mirko Hennig[3], Rumpa B. Bhattacharjee[3], Weining Yin[1], Hannah Golliher[1,4], Daniella Ishimaru[3], T. Noelle Lombana[3], David J. Lockhart[3], Brandon A. Wustman[3] and Lawrence E. Ostrowski[1,5,*]

## ABSTRACT

Primary ciliary dyskinesia (PCD) is a rare, genetically heterogenous disorder resulting from dysfunctional motile cilia that is characterized by chronic, progressive lung disease with currently no corrective therapies available. Here, we test the efficacy of selective organ targeting lipid nanoparticles (SORT-LNPs) that were optimized for potency and delivery to respiratory cells containing an mRNA encoding an axonemal protein to rescue ciliary activity in a murine culture model of PCD. Utilizing murine nasopharyngeal epithelial cell (mNPEC) cultures isolated from a conditional *Dnai1* knockout mouse model of the known human PCD-associated gene *DNAI1* homolog, we tested if SORT-LNPs containing an optimized *Dnai1* mRNA could rescue ciliary activity. Treatment of differentiating and well-differentiated *Dnai1* knockout mNPECs with SORT-LNP-*Dnai1* mRNA led to a dose-dependent increase in levels of DNAI1 protein and incorporation into ciliary axonemes, resulting in rescued ciliary activity with normal ciliary beat frequency that persisted for over 3 weeks. These data support further clinical development of an mRNA-based therapeutic with LNP-mediated delivery as a treatment for individuals with PCD with disease-causing *DNAI1* mutations.

KEY WORDS: Primary ciliary dyskinesia, Motile cilia, mRNA therapy, Lipid nanoparticles

## INTRODUCTION

Motile cilia are microtubule-based organelles that are essential for multiple processes involving the propulsion of extracellular fluid across the epithelium, including mucociliary clearance in the upper and lower respiratory tract (Spassky and Meunier, 2017; Wallmeier et al., 2020). Mutations in at least 50 different genes associated with the function of motile cilia result in a rare, genetically heterogeneous

disorder known as primary ciliary dyskinesia (PCD) (Horani and Ferkol, 2021; Knowles et al., 2016; Raidt et al., 2023). The exact prevalence of PCD is unknown; however, a recent study determined that the minimum worldwide occurrence is one in 7554 individuals, although this is likely to be an underestimation (Hannah et al., 2022). Due to impaired mucociliary clearance, PCD-affected individuals experience neonatal respiratory distress, chronic sinusitis, otitis media, frequent or chronic respiratory infections, and progressive lung damage leading to bronchiectasis or end-stage respiratory failure (Legendre et al., 2021; Leigh et al., 2019).

Current PCD therapies aim to maintain lung function and limit lung disease progression by improving airway clearance and reducing the severity and frequency of respiratory infections, as there are no corrective or disease-modifying therapies available (Paff et al., 2021; Rubbo and Lucas, 2017; Shapiro et al., 2016). The few published efforts to produce a corrective PCD treatment have used different modalities, including read-through therapy, gene therapy and mRNA therapy (Paff et al., 2021). Translational readthrough of nonsense mutations generated by aminoglycosides and non-aminoglycoside agents was evaluated; however, further identification of compounds that allow more efficient readthrough of premature stop codons without cell or long-term toxicity is required (Bukowy-Bieryłło et al., 2016; Dabrowski et al., 2021). Gene therapy strategies for developing a PCD therapeutic have included replacing or repairing the mutated gene sequences by utilizing adeno-associated viral vectors, lentiviral vectors and non-viral vectors for delivery of genetic material to airway epithelial cells (Bañuls et al., 2020; Chhin et al., 2009; Keiser et al., 2023; Lai et al., 2016; Ostrowski et al., 2014; Paff et al., 2021). While these gene therapy methods were effective for restoring ciliary activity *in vitro*, implementing these therapeutics *in vivo*, and ultimately in PCD-affected individuals, may prove challenging due to the small packaging capacity of viral vectors, the physical and immune barriers to apical vector delivery in the airways, and the ineffectiveness of repeat dosing due to development of stable neutralizing antibodies, as well as the maintenance of target gene expression without risking integrational mutagenesis effects (Bañuls et al., 2020; Keiser et al., 2023; Paff et al., 2021).

An alternative therapeutic approach under investigation utilizes an inhaled, non-viral delivery vector, a lipid nanoparticle (LNP), to deliver mRNA encoding the wild-type version of the mutated protein to target cells (Bañuls et al., 2020; Hou et al., 2021; Keiser et al., 2023; Paff et al., 2021). To promote cellular uptake of mRNA into cells and protect it from nuclease degradation and hydrolysis, mRNA is packaged in LNPs, spherical vesicles conventionally consisting of four lipid components, which, after cell delivery, undergo endosomal escape and release the mRNA cargo into the cytoplasm to be translated into protein. Compared to viral gene

[1]Marsico Lung Institute/Cystic Fibrosis Research and Treatment Center, University of North Carolina at Chapel Hill, Chapel Hill, NC 27599, USA. [2]Presently ReCode Therapeutics, Durham, NC 27701. [3]ReCode Therapeutics, Menlo Park, CA 94025, USA. [4]Department of Molecular Genetics and Microbiology, University of Florida, Gainesville, FL 32610, USA. [5]Department of Pediatrics, University of North Carolina at Chapel Hill, Chapel Hill, NC 27599, USA.

*Author for correspondence (ostro@med.unc.edu)

A.J.S., 0000-0002-3693-7917; M.H., 0009-0009-0238-737X; D.I., 0009-0009-3443-6869; L.E.O., 0000-0002-7558-6603

delivery systems, this method has several advantages, including the possibility of repeat administration with specific and efficient transduction of the lung epithelium, the ability to package mRNA encoding larger proteins, and the absence of viral integration. These properties make inhaled mRNA therapy an attractive option for clinical applications (Keiser et al., 2023; Hou et al., 2021). For example, there are multiple inhaled mRNA therapies utilizing LNP currently in clinical development for cystic fibrosis (Rowe et al., 2023; ClinicalTrials.gov ID NCT06237335).

Here, we explore the efficacy of using a specialized selective organ targeting lipid nanoparticle (SORT-LNP), which contains an additional fifth lipid, compared to conventional four component LNPs, and is designed to increase potency and delivery of mRNA to targeted lung epithelial cells, to rescue ciliary activity in a PCD *in vitro* model (Cheng et al., 2020; Sun et al., 2024; Wei et al., 2023). *Dnai1* knockout murine nasopharyngeal epithelial cell (mNPEC) cultures were used to test the efficacy of using SORT-LNP encapsulated *Dnai1* mRNA to restore ciliary activity (Cheng et al., 2020; Sun et al., 2024; Ostrowski et al., 2010; Wei et al., 2023; Zariwala et al., 2006). DNAI1 is an outer dynein arm (ODA) chain that is present along the length of the ciliary axoneme and is mutated in up to 10% of PCD cases (Ostrowski et al., 2010; Zariwala et al., 2006). DNAI1 comprises 701 amino acids with a molecular mass of ~80 kDa and has five conserved tryptophan-aspartate (WD) repeats involved in protein-protein interactions and ODA assembly. A conditional knockout mouse model to represent PCD was previously developed by using a tamoxifen-inducible Cre-lox system to delete exons 17 and 18 in *Dnai1*, encompassing 83 amino acids and one of the WD repeats, leading to a truncated and non-functional protein. This conditional deletion of *Dnai1* leads to the loss of ODA in the ciliary axonemes and immotile cilia, resulting in a PCD-like phenotype of severe rhinosinusitis in adult mice (Ostrowski et al., 2010).

In this study, basolateral treatment of mNPEC cultures at an air-liquid interface (ALI) with SORT-LNP containing *tdTomato* mRNA (SORT-LNP-*tdTomato*) demonstrated cellular uptake of SORT-LNP and expression of tdTomato fluorescence. Treatment of *Dnai1*$^{-/-}$ mNPEC cultures with SORT-LNP containing *Dnai1* mRNA (SORT-LNP-*Dnai1*) led to increased levels of wild-type DNAI1 protein and incorporation into ciliary axonemes, even in the presence of the mutant protein. The incorporation of exogenous wild-type DNAI1 resulted in the restoration of ciliary activity with a normal ciliary beat frequency that was maintained for several weeks in the absence of continued dosing. Rescued ciliary function was also shown to occur after treatment of *Dnai1*$^{-/-}$ mNPEC cultures with SORT-LNP-*Dnai1* in fully differentiated cultures. Thus, delivery of mRNA encapsulated in a lung-targeting SORT-LNP is a potential option for a corrective therapeutic for PCD and these results further support ongoing clinical development of this potential therapeutic.

## RESULTS

To test if SORT-LNP containing mRNA is a potential option for a disease-modifying treatment of PCD caused by mutations in *DNAI1* and to determine its efficacy for restoring ciliary function in a PCD model, airway epithelial cells derived from a tamoxifen-inducible *Dnai1* knockout mouse model were utilized. Nasopharynx tissue, due to the higher proportion of multiciliated cells present in the nasopharynx compared to the trachea (Woodworth et al., 2007), was isolated from *Dnai1*$^{flox/flox}$/*CreER*$^{+/-}$ mice, and mNPECs were expanded using conditionally reprogrammed cell (CRC) methods (Antunes et al., 2007; Eenjes et al., 2018; Suprynowicz et al., 2012;

You et al., 2002). A portion of these cells were treated with tamoxifen during this expansion phase, which induces the excision of exons 17 and 18 in *Dnai1* resulting in the loss of 88 amino acids and translation of a truncated, non-functional DNAI1 protein (Ostrowski et al., 2010). Nontreated control and tamoxifen-treated (*Dnai1*$^{-/-}$) cells were cultured at an ALI to allow for differentiation into a pseudostratified airway epithelium (Antunes et al., 2007; Eenjes et al., 2018; Suprynowicz et al., 2012; You et al., 2002). To maintain the air-liquid interface of the cultures, limit any effect of re-submersion of the cultures, including inhibition of ciliated cell differentiation, impairment of the cellular barrier integrity and alteration of cell signaling pathways (Clark et al., 1995; Gerovac et al., 2014; Mallek et al., 2024), and to maximize transfection of the LNPs, mNPEC cultures were treated basolaterally with the SORT-LNP-mRNA formulations in all studies. To formulate SORT-LNP mRNA, sequence-optimized mRNA transcripts encoding mouse *Dnai1* or human *DNAI1* were created by DNA-template-directed *in vitro* transcription of the mRNA using T7-RNA polymerase followed by 5′-capping, purification, resuspension and filtration. Purified mRNA was then loaded into SORT-LNP via nanoprecipitation, followed by buffer exchange, concentration and filtration (Wang et al., 2023). All LNP formulations were determined to have a particle size in the range of 62–118 nm, polydispersity index of 0.1, mRNA encapsulation efficiency in the range of 48–98% and mRNA concentration in the range of 0.5 to 1.08 mg/ml (Table S1).

First, to demonstrate uptake and expression of SORT-LNP encapsulated mRNA, differentiated mNPEC cultures were treated with 10 μg/ml SORT-LNP containing mRNA encoding the fluorescent protein tdTomato (SORT-LNP-*tdTomato* mRNA) basolaterally three times per week up to a total of five times and compared to untreated mNPEC cultures. Cultures were fixed 48 h after each treatment and then stained for nuclei and cilia. As expected, the untreated cultures showed no tdTomato protein expression, while treated cultures demonstrated high levels of tdTomato protein fluorescence (Fig. 1A). Quantification of the percentage of tdTomato-positive cells demonstrated that the level of tdTomato expression was similar between the mNPEC cultures that underwent one to five treatments, likely due to the tdTomato fluorescent signal being saturated after the first administration (Shaner et al., 2005) (Fig. 1B). We observed expression of tdTomato in multiciliated and non-ciliated cells, indicating that the SORT-LNP can efficiently transfect ciliated cells and target multiple cell types in differentiated mNPEC cultures.

To test if SORT-LNP encapsulating *Dnai1* mRNA can lead to expression of wild-type DNAI1 protein in *Dnai1*$^{-/-}$ mNPECs, differentiating cultures were treated with SORT-LNP containing untagged *Dnai1* mRNA or hemagglutinin (HA)-tagged *Dnai1* mRNA. Starting at ALI day 3–4, *Dnai1*$^{-/-}$ cultures were treated with 10 μg/ml of SORT-LNP-*Dnai1* or SORT-LNP-*Dnai1-HA* basolaterally three times per week for a total of four treatments. After cultures reached ALI day 19, mNPECs were lysed and immunoblots on whole-cell lysates of control and treated mNPEC cultures were performed probing for DNAI1 and HA-tag. In the control *Dnai1*$^{flox/flox}$ mNPEC cultures that were not treated with tamoxifen, a strong band at 80 kDa was detected representing wild-type DNAI1 expression, while in the *Dnai1*$^{-/-}$ mNPEC cultures treated with tamoxifen, expression of wild-type DNAI1 is replaced with the expression of the truncated nonfunctional DNAI1 at 70 kDa. In the *Dnai1*$^{-/-}$ mNPEC cultures that were also treated with SORT-LNP-*Dnai1* (*Dnai1*$^{-/-}$ LNP-*Dnai1*) or SORT-LNP-*Dnai1-HA* (*Dnai1*$^{-/-}$ LNP-*Dnai1-HA*), there was an increase in the

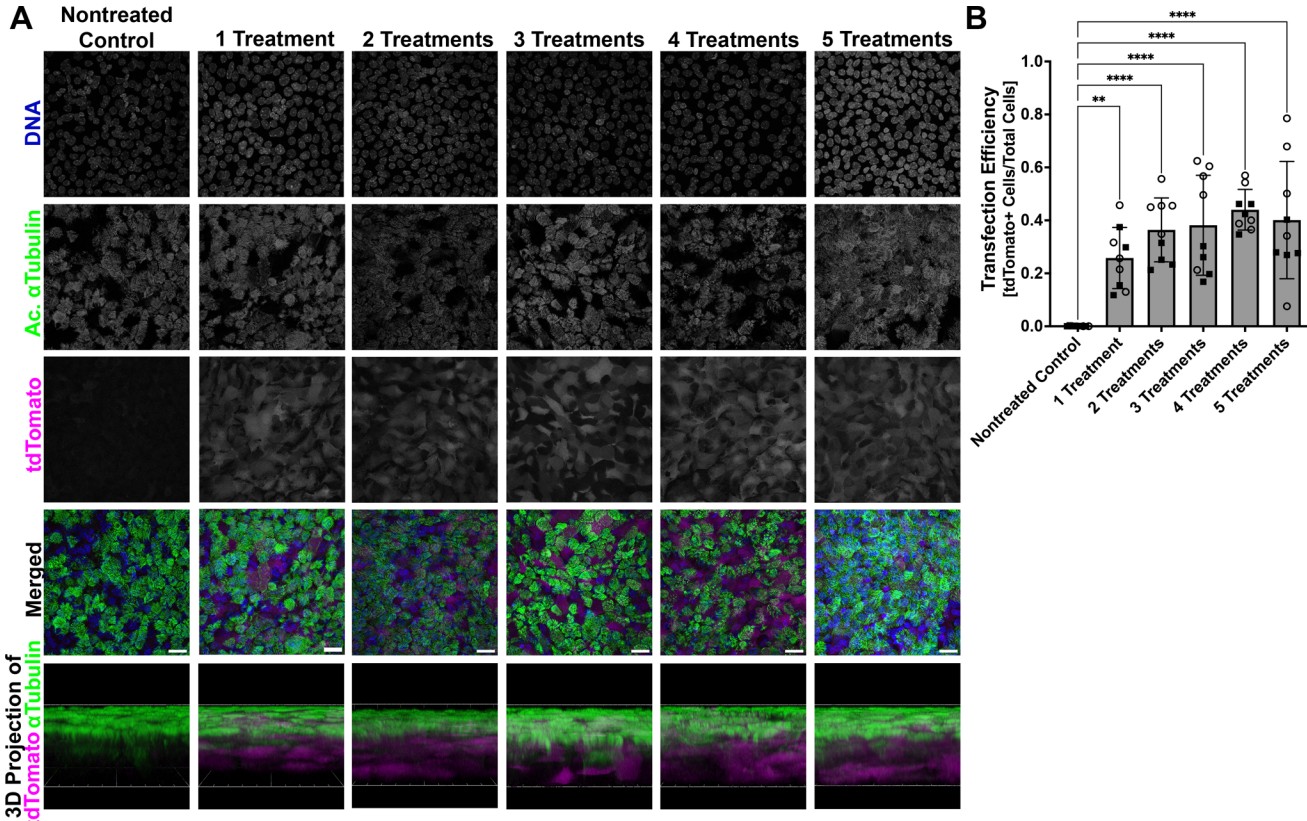

**Fig. 1. Treatment of mNPEC cultures with SORT-LNP-*tdTomato* demonstrates robust cellular uptake and tdTomato expression.** (A) Representative whole-mount immunofluorescence images of differentiated wild-type murine nasopharyngeal epithelial cell (mNPEC) cultures either not treated (nontreated control) or treated with 10 µg/ml SORT-LNP-*tdTomato* mRNA three times per week up to a total of five times starting at ALI day 19 showing tdTomato (magenta), acetylated α-tubulin (green) and DNA (blue) fluorescence. Images are representative of a total of 10 fields at 40× magnification across two independent experiments. Scale bars: 20 µm. Scale grid for 3D projections: 40 µm. (B) Transfection efficiency of SORT-LNP-*tdTomato* mRNA represented by quantification of the total number of tdTomato⁺ cells over the total number of nuclei in immunofluorescent images of SORT-LNP-*tdTomato*-treated mNPEC cultures. Data are mean±s.d., *n*=10 fields of view across two independent experiments. Open circle and square datapoints represent fields of view from the first and second replicate experiment, respectively. A one-way ordinary ANOVA followed by Tukey's multiple comparison test determined a significant difference between the means of treated mNPEC cultures and the nontreated control (**$P<0.01$; ****$P<0.0001$).

expression of wild-type DNAI1 and clear expression of HA-tagged DNAI1, respectively (Fig. 2A). Quantification of the wild-type DNAI1 protein levels detected by immunoblots indicates that four treatments of SORT-LNP-*Dnai1* was sufficient to restore 40.7±7.2% of control DNAI1 protein levels, while SORT-LNP-*Dnai1-HA* was able to restore DNAI1 expression to approximately 14.5±11.5% (mean±s.d.) (Fig. 2B).

To establish if the exogenous DNAI1 protein can maintain its function of forming ODAs and be incorporated into ciliary axonemes, whole-mount immunocytochemistry labeling nuclei, cilia and the HA-tag was performed on control and treated mNPEC cultures. Immunostaining for the HA-tag was performed to clearly distinguish between exogenous DNAI1 and any residual endogenous DNAI1. Except for background signal, no expression of HA-tag was detected in control *Dnai1*^flox/flox^ cultures, *Dnai1*^−/−^ cultures or *Dnai1*^−/−^ cultures treated with SORT-LNP-*Dnai1*. However, expression of the HA-tag was detected in ~38.6±14.8% (mean±s.d.) of cells in *Dnai1*^−/−^ cultures treated with SORT-LNP-*Dnai1-HA*, where the HA-tag was localized to the ciliary axonemes (Fig. 2C,D), suggesting that treatment with SORT-LNP containing *Dnai1* or *Dnai1-HA* mRNA can lead to expression of exogenous wild-type DNAI1 that can be incorporated into ciliary axonemes. Although the HA-tag present on the C-terminus of DNAI1 does decrease its expression levels, it clearly demonstrates the expression

and axonemal incorporation of exogenous DNAI1 protein. It is interesting to note that this incorporation took place in the presence of the endogenous truncated DNAI1 protein, suggesting that, at least in this model, the mutant protein did not exert a dominant-negative effect.

Having demonstrated that the SORT-LNP-*Dnai1* was able to express DNAI1 that was incorporated into ciliary axonemes, high-speed video microscopy was used to determine if treatment with SORT-LNP-*Dnai1* could restore ciliary activity in *Dnai1*^−/−^ mNPEC cultures (Sears et al., 2015; Sisson et al., 2003). *Dnai1*^−/−^ mNPEC cultures were treated with 10 µg/ml SORT-LNP-*Dnai1* basolaterally during differentiation three times per week for a total of four times starting at ALI day 3. The active area, or the percentage of surface area that contains active or beating cilia, of cultures was measured once every 7 days during and after differentiation up to ALI day 35. Control *Dnai1*^flox/flox^ mNPEC cultures reached a maximum mean active area of 29.5±2.2% (mean±s.d.) by ALI day 14, followed by a decrease and plateauing throughout the maintenance of differentiated cultures to a mean active area of 25.4±1.7% by ALI day 35. *Dnai1*^−/−^ mNPEC cultures reached a maximum mean active area of 1.4±0.3% (4.8% of active area in control cultures) by ALI day 35. However, *Dnai1*^−/−^ mNPEC cultures that were treated with SORT-LNP-*Dnai1* reached a maximum mean active area of 26.2±3.0% (89% of active area in

Journal of Cell Science

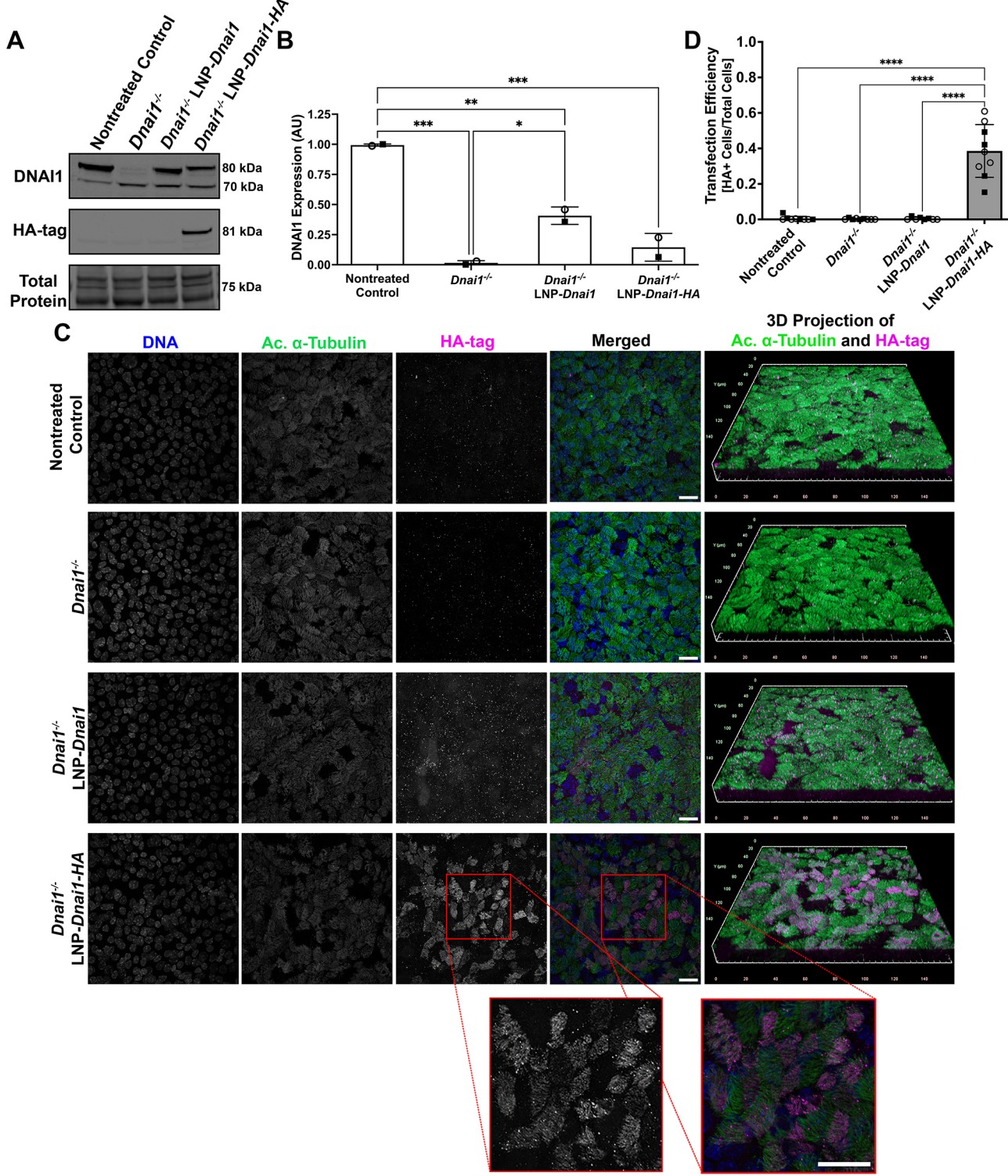

**Fig. 2.** See next page for legend.

control cultures) by ALI day 14, followed by a similar pattern to control cultures with a decrease and plateauing at ~14.4±2.3% active area (57% of active area in control cultures) by ALI day 35 (Fig. 3A, Fig. S1 and Movies 1–3). Importantly, ciliary activity in the SORT-LNP-*Dnai1*-treated cultures was maintained at near control levels for over 3 weeks after the last treatment. The mean active areas of the *Dnai1*$^{-/-}$ mNPEC cultures and the SORT-LNP-*Dnai1*-treated *Dnai1*$^{-/-}$ mNPEC cultures were significantly different from one another ($P<0.0001$), while the mean active areas of control *Dnai1*$^{flox/flox}$ mNPEC cultures and SORT-LNP-*Dnai1*-treated *Dnai1*$^{-/-}$ mNPEC cultures were not significantly different ($P=0.22$) on ALI day 14. Thus, treatment of *Dnai1*$^{-/-}$ mNPEC cultures with SORT-LNP-*Dnai1* can restore ciliary motility approaching the level of control mNPEC cultures.

**Fig. 2. SORT-LNP-*Dnai1* treatment of *Dnai1*$^{-/-}$ mNPEC cultures leads to wild-type DNAI1 expression and incorporation into ciliary axonemes.** Differentiating *Dnai1*$^{-/-}$ murine nasopharyngeal epithelial cell (mNPEC) cultures were treated with SORT-LNP formulations at 10 µg/ml three times per week for four treatments starting at ALI day 3. (A) Representative immunoblot probing for DNAI1 and HA-tag on whole-cell lysates of *Dnai1*$^{flox/flox}$ nontreated control and *Dnai1*$^{-/-}$ mNPEC cultures collected 10 days after the last treatment. (B) Quantification of wild-type (80 kDa) DNAI1 protein levels in immunoblots from A. Raw signals were normalized to a total protein stain, and control ratios were set to 1.0. Data are mean±s.d. across two independent experiments. (C) Representative whole-mount immunofluorescence images of *Dnai1*$^{flox/flox}$ and *Dnai1*$^{-/-}$ mNPEC cultures fixed at and stained for DNA (blue) using Hoechst33342, HA-tag (magenta) and acetylated α-tubulin (green) to mark ciliary axonemes. mNPEC cultures were fixed at ALI day 20, ~10 days after the last treatment with SORT-LNP-mRNA. The red square outlines a zoomed in section highlighting incorporation of the HA-tag in ciliary axonemes. Images are representative of a total of 10 fields at 40× magnification across two independent experiments. Scale bars: 20 µm. Scale grid for 3D projections: 150 µm. (D) Transfection efficiency of SORT-LNP-*Dnai1-HA* mRNA represented by quantification of the total number of HA-tag$^+$ cells over the total number of nuclei in immunofluorescent images of SORT-LNP-mRNA-treated mNPEC cultures. Data are mean±s.d., *n*=10 fields of view across two independent experiments. Open circles and squares represent fields of view from the first and second replicate experiment, respectively. For B and D, a one-way ordinary ANOVA followed by Tukey's multiple comparison test determined a significant difference between the means of treated mNPEC cultures and the nontreated control (**P*<0.05; ***P*<0.01; ****P*<0.001; *****P*<0.0001).

Additionally, treatment of *Dnai1*$^{-/-}$ mNPEC cultures with SORT-LNP containing HA-tagged *Dnai1* mRNA were also able to rescue ciliary activity, albeit at lower levels compared to SORT-LNP containing untagged *Dnai1* mRNA (Fig. S2A,B).

To determine if the rescued ciliary activity induced by SORT-LNP-*Dnai1* treatment reaches a normal ciliary beat frequency (CBF), high-speed video microscopy was also used to measure CBF of control and treated *Dnai1*$^{-/-}$ mNPEC cultures. CBF measurements were performed on control *Dnai1*$^{flox/flox}$, *Dnai1*$^{-/-}$ and *Dnai1*$^{-/-}$ mNPEC cultures treated with SORT-LNP-*Dnai1*. Cultures were studied on ALI day 35 or 39, ~3 weeks after the last treatment. Control *Dnai1*$^{flox/flox}$ mNPEC cultures had a mean CBF of 17.1±2.2 Hz, while *Dnai1*$^{-/-}$ mNPEC cultures had an average CBF of 9.7±5.1 Hz (mean±s.d.; Fig. 3B). The detection of ciliary activity in the *Dnai1*$^{-/-}$ mNPEC cultures is attributed to a small number of multiciliated cells with active cilia in a few of the fields of view, probably due to incomplete *Dnai1* deletion in these cells (Fig. 3C; Movies 1–3). Interestingly, *Dnai1*$^{-/-}$ mNPEC cultures treated with SORT-LNP-*Dnai1* had a mean CBF of 14.5±3.0 Hz, which was not statistically different from the mean CBF of the control *Dnai1*$^{flox/flox}$ mNPEC cultures (*P*=0.44) (Fig. 3B). Representative heat maps of fields of view demonstrate that the rescued cilia of the *Dnai1*$^{-/-}$ mNPEC cultures treated with SORT-LNP-*Dnai1* reach a similar maximum CBF compared to control cultures (Fig. 3C; Movies 1–3). The heat maps of these cultures also illustrate that the detection of CBF in *Dnai1*$^{-/-}$ mNPEC cultures is due to a few active multiciliated cells with incomplete *Dnai1*$^{-/-}$ deletion. *Dnai1*$^{-/-}$ mNPEC cultures treated with SORT-LNP containing HA-tagged *Dnai1* mRNA were also able to recover a mean CBF within a normal range (Rogers et al., 2022) (Fig. S2C). Therefore, treatment of *Dnai1*$^{-/-}$ mNPEC cultures with SORT-LNP-*Dnai1* can restore ciliary activity to normal CBF levels. Additionally, high-resolution videos of motile cilia from the SORT-LNP-*Dnai1* treated *Dnai1*$^{-/-}$ mNPEC cultures suggest that rescued

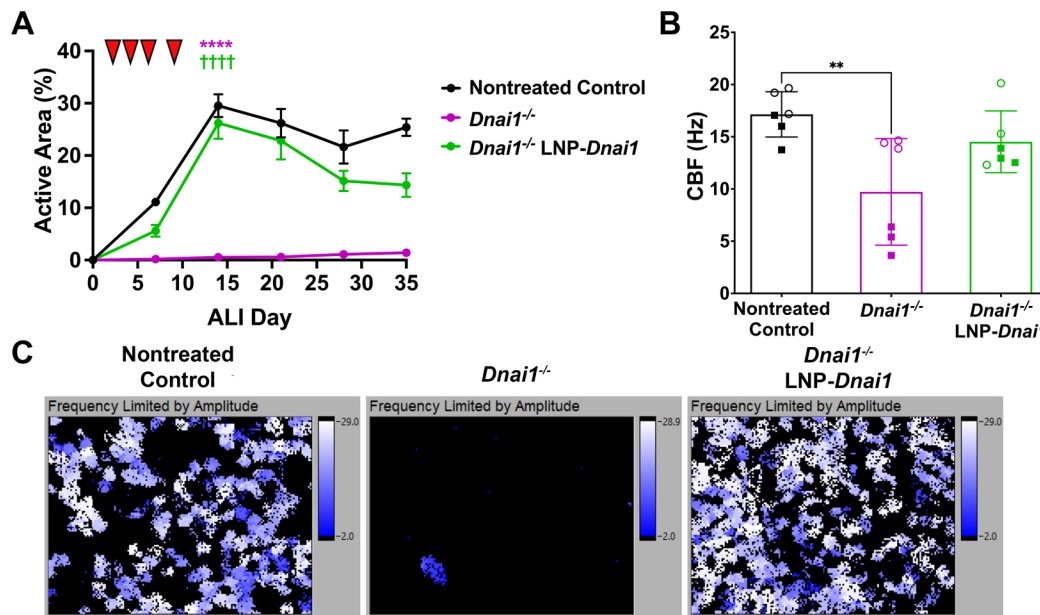

**Fig. 3. SORT-LNP-*Dnai1* treatment rescues and maintains ciliary activity with normal ciliary beat frequency in *Dnai1*$^{-/-}$ mNPEC cultures.** Differentiating *Dnai1*$^{-/-}$ murine nasopharyngeal epithelial cell (mNPEC) cultures were treated basolaterally with 10 µg/ml SORT-LNP-*Dnai1* on ALI days 3, 5, 7 and 10 (red arrowheads in A) for a total of four times. (A) Active area measurements taken on ALI days 7, 14, 21, 28 and 35 were performed in triplicate and are shown as mean±s.d., with replicate data presented in Fig. S1. (B) Ciliary beat frequency (CBF) measurements on ALI day 35 or 39 are shown as mean±s.d. from two independent experiments performed in triplicate. Open circle and square datapoints represent mean CBF from individual cultures from the first and second replicate experiment, respectively. (C) Heat maps showing range of ciliary beat frequencies from whole field of views. Heat maps correspond to Movies 1–3 showing ciliary activity of whole field of views taken using SAVA software on a Nikon Eclipse TE2000 inverted microscope with a 20×objective. For A and B, a one-way ANOVA followed by a Tukey's multiple comparisons test determined that the mean active areas on ALI day 14 were significantly different when compared to nontreated control (*****P*<0.0001) and when compared to *Dnai1*$^{-/-}$ (††††*P*<0.0001), while the mean CBF was significantly different when compared to nontreated control (***P*<0.01).

Journal of Cell Science

cilia potentially follow a normal waveform with some level of coordination (Movies 4–9).

Next, dose-response studies were performed to determine the optimal dosage regimen to provide maximum rescued ciliary activity. Immunoblots examining DNAI1 protein levels and active area measurements were performed on differentiating $Dnai1^{-/-}$ mNPEC cultures treated with 1, 5, 10 or 20 µg/ml of SORT-LNP-$Dnai1$ three times week for a total of four times. A typical dose response was observed both with DNAI1 protein levels detected by immunoblots and active area measurements, where the maximum mean active areas reached for 1, 5, 10 and 20 µg/ml concentrations were 0.23±0.1%, 17.6±2.6%, 30.7±3.1% and 37.3±1.7% (mean±s.d.), respectively, by ALI day 21. These mean active areas corresponded to 1%, 41%, 71% and 86%, respectively, of the active area of control mNPEC cultures, which had a maximum mean active area of 43.2±1.2%. While the 20 µg/ml dose produced the highest active areas, these treated cultures exhibited traits of loss of cell culture integrity, including cell death, culture leakiness and gaps in the epithelial layer. The 10 µg/ml dose showed minimal signs of loss of cell culture integrity, which included the inability to maintain an air-liquid interface (i.e. leakiness). Therefore, the optimal dose concentration at a three times per week dosing frequency was determined to be 5 µg/ml, which provided the maximum ciliary active area without any loss of culture quality (Fig. 4A–C; Fig. S3A). Immunoblots on whole cell lysates from differentiating $Dnai1^{-/-}$ mNPEC cultures treated with 5 µg/ml of SORT-LNP-$Dnai1$ three times per week for a total of two (2×), four (4×), six (6×) or eight (8×) times showed that wild-type DNAI1 protein levels reached the highest level at 56.7±37.9% of control mNPEC culture levels when treated with 5 µg/ml of SORT-LNP-$Dnai1$ three times per week for a total of six times. Active area measurements again demonstrated a dose response corresponding with increasing number of treatments. The maximum ciliary active area means for the $Dnai1^{-/-}$ LNP-$Dnai1$ 2×, 4×, 6× and 8× experimental groups were 10.1±1.0%, 24.0±0.5%, 28.5±1.0% and 28.6±1.1%, respectively, which corresponded with 21%, 50%, 58% and 58% of the active area of control mNPEC cultures with a maximum ciliary active area mean of 48.9±1.3%. Treatment with 5 µg/ml of SORT-LNP-$Dnai1$ for a total of six times at a dosing frequency of three times per week produced the maximum ciliary active area without loss of cell culture integrity (Fig. 4D–F; Fig. S3B). Thus, a treatment regimen with a 5 µg/ml dose concentration given three times per week for a total of six treatments was determined to produce the optimal levels of ciliary rescue in differentiating mNPEC cultures undergoing ciliogenesis.

A significant obstacle to developing an effective, corrective treatment for PCD is the difficulty of transducing nucleic material into non-dividing, terminally differentiated airway cells (Bals et al., 1999; Pickles et al., 1998). To explore if SORT-LNP containing $Dnai1$ mRNA could restore ciliary function in differentiated mNPEC cultures, cultures were treated with 5 µg/ml of SORT-LNP-$Dnai1$ or SORT-LNP-$Dnai1$-HA three times per week for a total of six treatments starting at ALI day 17–20, when mNPECs were considered well differentiated. $Dnai1^{-/-}$ mNPEC cultures were additionally treated with SORT-LNP containing $tdTomato$ mRNA as a vehicle control. Immunoblots for DNAI1 on whole-cell lysates from differentiated $Dnai1^{-/-}$ mNPEC cultures showed a 29% and a 10% increase in DNAI1 protein levels when treated with SORT-LNP-$Dnai1$ and SORT-LNP-$Dnai1$-HA, respectively, when compared to untreated $Dnai1^{-/-}$ mNPEC cultures (Fig. 5A,B). Baseline ciliary active area measurements taken on ALI day 17 showed that control $Dnai1^{flox/flox}$ mNPEC cultures had an active area

of 40.0±0.3%, while the untreated $Dnai1^{-/-}$ mNPEC cultures had an active area of ~0.1±0.03%. Remarkably, within 7 days of the start of SORT-LNP-mRNA treatment, the active area of SORT-LNP-$Dnai1$ or SORT-LNP-$Dnai1$-HA treated $Dnai1^{-/-}$ mNPEC cultures had increased to 11.7±0.3% and 5.0±0.9% (32% and 14% of the active area of control mNPEC cultures), respectively. SORT-LNP-$Dnai1$-treated $Dnai1^{-/-}$ mNPEC cultures eventually reached a maximum mean active area of 25.3±1.0% (65% of the active area of control cultures) by ALI day 38, and SORT-LNP-$Dnai1$-HA treated $Dnai1^{-/-}$ mNPEC cultures reached a maximum mean active area of 15.1±3.7% (42% of the active area of control cultures) by ALI day 31. The untreated $Dnai1^{-/-}$ mNPEC cultures and SORT-LNP-$tdTomato$-treated $Dnai1^{-/-}$ mNPEC cultures showed no significant ciliary activity, with a maximum mean active area of 0.2±0.1% on ALI day 38 (Fig. 5C; Fig. S4). Both mean ciliary active areas of the $Dnai1^{-/-}$ mNPEC cultures treated with SORT-LNP-$Dnai1$ and SORT-LNP-$Dnai1$-HA ($P<0.0001$), but not SORT-LNP-$tdTomato$ ($P>0.9999$), were significantly different compared to the untreated $Dnai1^{-/-}$ mNPEC cultures on ALI day 31. CBF measurements on ALI day 38 or 40, while significantly decreased from nontreated controls, demonstrated restoration of mean CBF to within a normal range when differentiated $Dnai1^{-/-}$ mNPEC cultures were treated with SORT-LNP-$Dnai1$ or SORT-LNP-$Dnai1$-HA (Rogers et al., 2022) (Fig. 5D). In summary, treatment with SORT-LNP encapsulating $Dnai1$ mRNA can lead to the expression of exogenous wild-type DNAI1 protein that restore ciliary activity with a normal CBF, even in well-differentiated $Dnai1^{-/-}$ mNPEC cultures.

Finally, we tested the ability of human $DNAI1$ mRNA to rescue ciliary activity in differentiating $Dnai1^{-/-}$ mouse cultures. We hypothesized that the high similarity between these protein homologs (89%) may enable us to detect ciliary rescue across species. $Dnai1^{-/-}$ mNPEC cultures were treated with 5 µg/ml of either SORT-LNP-$tdTomato$ as a vehicle control, SORT-LNP-$Dnai1$ or SORT-LNP-$DNAI1$ three times per week for a total of six times starting on ALI day 2. Immunoblots detecting DNAI1 protein from whole cell lysates of untreated and treated $Dnai1^{-/-}$ mNPEC cultures showed rescued expression of wild-type DNAI1 in cultures treated with SORT-LNP-$Dnai1$ that reached a level of 27.3±0.2% (mean±s.d.) of control $Dnai1^{flox/flox}$ mNPEC cultures. In the $Dnai1^{-/-}$ mNPEC cultures treated with SORT-LNP-$DNAI1$, there was a shift in the molecular mass of the detected wild-type protein corresponding to the human DNAI1 protein relative to the mouse protein (Woo et al., 2022) (Fig. 6A,B). Active area measurements of the control $Dnai1^{flox/flox}$ mNPEC cultures reached a maximum of 36.3±0.3% by ALI day 16, while the $Dnai1^{-/-}$ and the SORT-LNP-$tdTomato$-treated $Dnai1^{-/-}$ mNPEC cultures only reached a maximum active area of 0.2±0.1% and 0.2±0.2%, respectively, by ALI day 23. Intriguingly, the $Dnai1^{-/-}$ mNPEC cultures treated with SORT-LNP-$DNAI1$ or SORT-LNP-$Dnai1$ reached similar maximum mean active areas of 28.9±3.7% and 29.6±2.0% (80% and 82% of the active area of control cultures), respectively, by ALI day 16 (Fig. 6C; Fig. S5). Both mean ciliary active areas of the $Dnai1^{-/-}$ mNPEC cultures treated with SORT-LNP-$Dnai1$ and SORT-LNP-$DNAI1$ ($P<0.0001$), but not SORT-LNP-$tdTomato$ ($P>0.9999$), were significantly different compared to the untreated $Dnai1^{-/-}$ mNPEC cultures on ALI day 16. CBF measurements of rescued ciliary activity on ALI day 23 or 27 demonstrated that the average CBF increased in $Dnai1^{-/-}$ mNPEC cultures treated with SORT-LNP-$Dnai1$ or SORT-LNP-$DNAI1$ compared to $Dnai1^{-/-}$ or SORT-LNP-$tdTomato$-treated cultures (Fig. 6D). Thus, human $DNAI1$ mRNA encapsulated in SORT-LNP has the capacity to

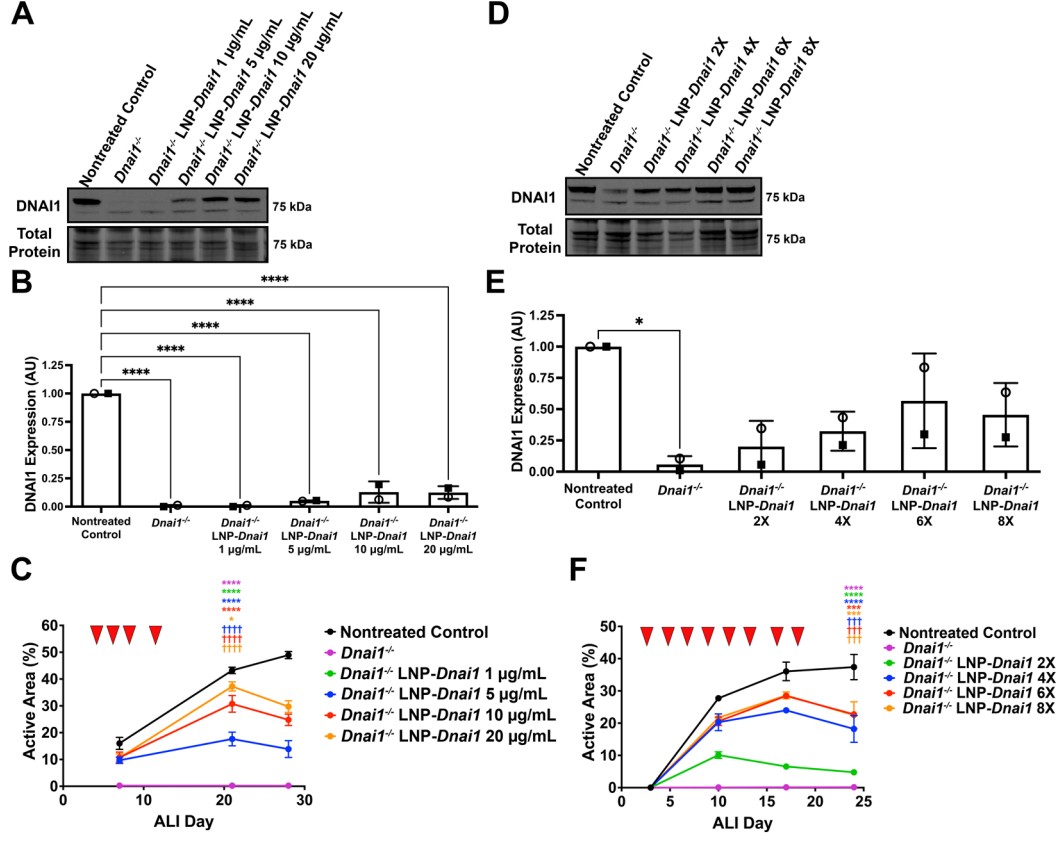

**Fig. 4. Six treatments of SORT-LNP-*Dnai1* at 5 µg/ml produce optimal rescued ciliary activity levels.** (A–C) Differentiating *Dnai1*$^{-/-}$ murine nasopharyngeal epithelial cell (mNPEC) cultures were treated basolaterally three times per week with SORT-LNP-*Dnai1* mRNA at concentrations of 1, 5, 10 or 20 µg/ml for a total of four times on ALI days 4, 6, 8 and 11 (red arrowheads in C). (D–F) Differentiating *Dnai1*$^{-/-}$ mNPEC cultures were treated basolaterally three times per week with SORT-LNP-*Dnai1* mRNA at the optimal 5 µg/ml concentration for a total of 2, 4, 6 or 8 treatments that were given on ALI days 2, 4, 6, 9, 11, 13, 16 and 18 (red arrowheads in F). (A) Representative immunoblot probing for DNAI1 on whole-cell lysates from *Dnai1*$^{flox/flox}$ and *Dnai1*$^{-/-}$ mNPEC cultures comparing SORT-LNP-*Dnai1* dosing concentrations. (B) Quantification of wild-type (80 kDa) DNAI1 protein levels detected with immunoblots from A. Raw signals were normalized to total protein stain, then control ratios were set to 1.0. Data are mean±s.d. from two independent experiments. (C) Active area measurements taken at ALI days 7, 21 and 28 were performed in triplicate and are shown as mean±s.d., with replicate data represented in Fig. S3A. (D) Representative immunoblot probing for DNAI1 on whole-cell lysates from *Dnai1*$^{flox/flox}$ and *Dnai1*$^{-/-}$ mNPEC cultures comparing numbers of SORT-LNP-*Dnai1* treatments at a 5 µg/ml dose. (E) Quantification of wild-type (80 kDa) DNAI1 protein levels detected with immunoblots from D. Raw signals were normalized to total protein stain, then control ratios were set to 1.0. Data are mean±s.d. from two independent experiments. (F) Active area measurements taken at ALI days 3, 10, 17 and 24 were performed in triplicate and are shown as mean±s.d., with replicate data presented in Fig. S3B. For B and E, a one-way ordinary ANOVA followed by Tukey's multiple comparison test determined a significant difference between the means when compared to the nontreated control (*$P<0.05$; ****$P<0.0001$). For C and F, the mean active areas on ALI day 21 or ALI day 24 of the *Dnai1*$^{-/-}$ LNP-*Dnai1* groups were significantly different from the nontreated control group (*$P<0.05$; ***$P<0.001$; ****$P<0.0001$) and the *Dnai1*$^{-/-}$ group ($^{†††}P<0.001$; $^{††††}P<0.0001$) based on a one-way ANOVA followed by a Tukey's multiple comparisons test.

express functional wild-type DNAI1 protein that can restore ciliary activity in murine PCD cells as effectively as its mouse homolog.

## DISCUSSION

Primary ciliary dyskinesia is a rare, genetically heterogenous disease in which mutations in over 50 different genes cause defects in motile cilia, resulting in impairment of ciliary motility and multiple cilia-associated processes, including mucociliary clearance. As a result of impaired mucociliary clearance, individuals with PCD suffer from chronic upper and lower respiratory infections, and progressive lung disease that leads to bronchiectasis and, in severe cases, requires a lung transplant (Leigh et al., 2019). Currently, there are no corrective therapies for PCD. Because of the genetic heterogeneity of the disease, it is likely that a personalized medicine approach will be required to correct the underlying defect (Keiser et al., 2023). SORT-LNPs that have been designed to increase potency and delivery of an mRNA encoding

the wild-type protein required for functional motile cilia to targeted respiratory epithelial cells have the potential to restore ciliary activity and improve the disease phenotype. To restore ciliary activity, a sufficient amount of translated and functional protein must be incorporated into and throughout the length of the ciliary axoneme. Thus, the key objectives of this work were to confirm that SORT-LNPs containing *Dnai1* mRNA can be efficiently delivered to and transfect *Dnai1*$^{-/-}$ airway epithelial cells, to evaluate the extent to which delivered and translated *Dnai1* mRNA can restore DNAI1 protein levels and functional ciliary activity, and to determine the durability of the restored ciliary activity.

First, using SORT-LNPs containing fluorescent reporter *tdTomato* mRNA, treatment of mNPEC cultures demonstrated uptake and expression of the mRNA cargo in ciliated and non-ciliated airway epithelial cells. Because almost all cells are in contact with the basement membrane in the pseudostratified airway epithelium present in mNPEC cultures, basolateral treatment allows

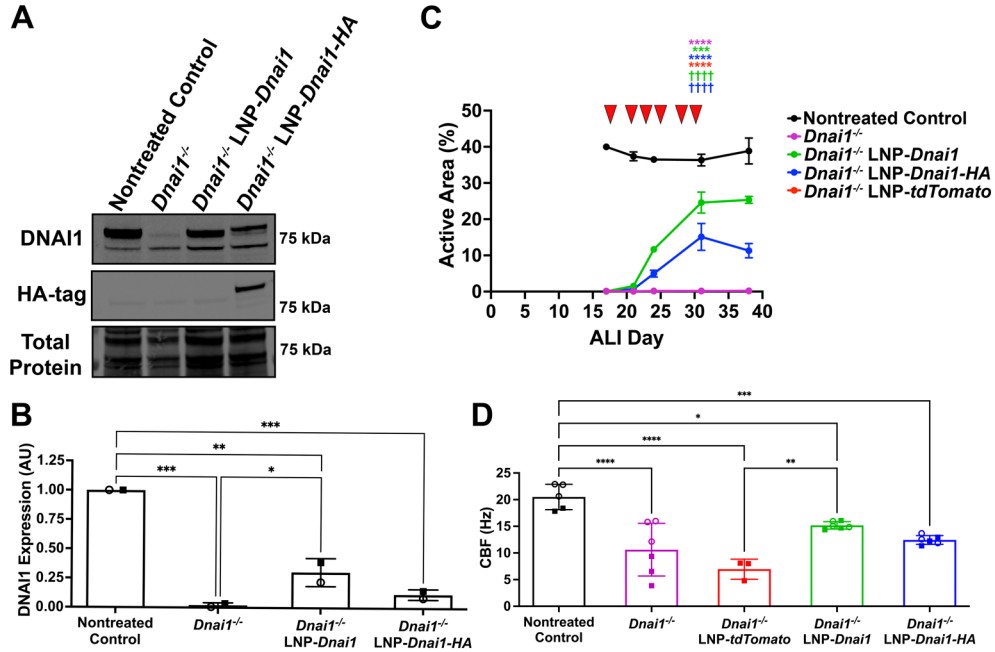

**Fig. 5. Treatment with SORT-LNP-*Dnai1* rescues ciliary activity in fully differentiated *Dnai1*⁻/⁻ mNPEC cultures.** Fully differentiated *Dnai1*⁻/⁻ murine nasopharyngeal epithelial cell (mNPEC) cultures were treated with 5 µg/ml of SORT-LNP-mRNA basolaterally three times per week for a total of six treatments on ALI days 18, 21, 23, 25, 28 and 30 (red arrowheads in C). (A) Representative immunoblot probing for DNAI1 and HA-tag on whole-cell lysates from *Dnai1*ᶠˡᵒˣ/ᶠˡᵒˣ and *Dnai1*⁻/⁻ mNPEC cultures. (B) Quantification of wild-type (80 kDa) DNAI1 protein levels detected in immunoblots represented in A. Raw signals were normalized to total protein stain, then control ratios were set to 1.0. Data are mean±s.d. from two independent experiments. (C) Active area measurements taken on ALI days 17, 21, 24, 31 and 38 were performed in triplicate and are shown as mean±s.d, with replicate data presented in Fig. S4. A one-way ANOVA with a Tukey's multiple comparisons test determined that the mean active areas on ALI day 31 of the treated mNPEC cultures were significantly different when compared to the nontreated control cultures (***$P<0.001$, ****$P<0.0001$) and when compared to the *Dnai1*⁻/⁻ cultures (††††$P<0.0001$). (D) Ciliary beat frequency (CBF) measurements on ALI day 38 or 40. Data shown are mean±s.d. from two independent experiments performed in triplicate. Open circle and square datapoints represent mean CBF from individual cultures from the first and second replicate experiment, respectively. For B and D, an ordinary one-way ANOVA test with Tukey's multiple comparisons determined a significant difference between the mean DNAI1 expression and mean CBFs (*$P<0.05$; **$P<0.01$; ***$P<0.001$; ****$P<0.0001$).

all cells to be transfected directly. Approximately 40% of cells of mNPEC cultures were determined to be transfected by SORT-LNP-*tdTomato* mRNA through basolateral delivery (Fig. 1B). Hennig et al. (2025) determined, using flow cytometry of human bronchial epithelial cells transfected with SORT-LNP-mRNA through basolateral or nebulized apical delivery, that the cell-type distribution of transfected cells includes ∼90% basal cells and 10% ciliated cells with basolateral delivery, but more ciliated and secretory cells with apical delivery, which suggests that the majority of cells transfected in mNPEC cultures may have been basal cells. Further evaluation is required to confirm the uptake and cell biodistribution of SORT-LNPs in mNPEC cultures and how it is influenced by the method of delivery.

Second, when treated with SORT-LNPs containing *Dnai1* mRNA, *Dnai1*⁻/⁻ mNPEC cultures expressed an exogenous, wild-type DNAI1 protein that is incorporated into the ciliary axoneme. The presence of a functional, exogenous DNAI1 protein in these mNPEC cultures resulted in rescued ciliary activity that reached comparable activity levels measured in control *Dnai1*ᶠˡᵒˣ/ᶠˡᵒˣ mNPEC cultures. The mean CBF of *Dnai1*⁻/⁻ mNPEC cultures increased after treatment with SORT-LNP encapsulated *Dnai1*, *Dnai1-HA* or *DNAI1* mRNA, as did the percentage of ciliated cells with active cilia. The rescued motile cilia had a mean CBF in the normal range of 8 to 18 Hz, but did not return to Dnai1ᶠˡᵒˣ/ᶠˡᵒˣ CBF levels in some cases (Figs 5D and 6D) (Rogers et al., 2022). The lower CBF levels in treated *DNAI1*⁻/⁻ mNPEC cultures could indicate that there was sufficient DNAI1 protein incorporated into

the cilia to provide the force for motility to reach the lower range of normal CBF levels, but not to reach the levels required for rescuing CBF completely to *Dnai1*ᶠˡᵒˣ/ᶠˡᵒˣ levels. Further analysis is required to correlate the amount of incorporated DNAI1 protein in rescued ciliary axonemes to the level of CBF achieved, if this has an impact on ciliary waveform and mucociliary transport, and if it is clinically meaningful. Moreover, high-resolution videos of ciliary activity from these SORT-LNP-*Dnai1* mRNA-treated *Dnai1*⁻/⁻ mNPEC cultures suggest that the rescued cilia have a normal waveform pattern and are coordinated with each other. Further analysis is required to determine if the extent of this coordination is sufficient to have a significant impact on mucociliary transport.

Third, following treatment of *Dnai1*⁻/⁻ mNPEC cultures with SORT-LNP-*Dnai1* mRNA, rescued ciliary activity reached its maximum active area from 2–10 days after the last dose, dependent on the treatment regimen. The rescued ciliary activity was observed to be persistent, as these cultures maintained at least 50% of their maximum mean active area levels up to 28 days after the last dose, which is the last day active areas were measured, indicating that the duration of rescued ciliary activity persists for at least 1 month, if not longer. Additionally, this suggests that the newly made exogenous DNAI1 protein produced is very stable, most likely due to incorporation into the ciliary axoneme and interactions with its ODA counterparts. Based on these results, and due to the highly conserved nature of mouse and human cilia, an mRNA therapy that leads to the expression of a human ciliary structural protein, such as DNAI1, is expected to produce a protein

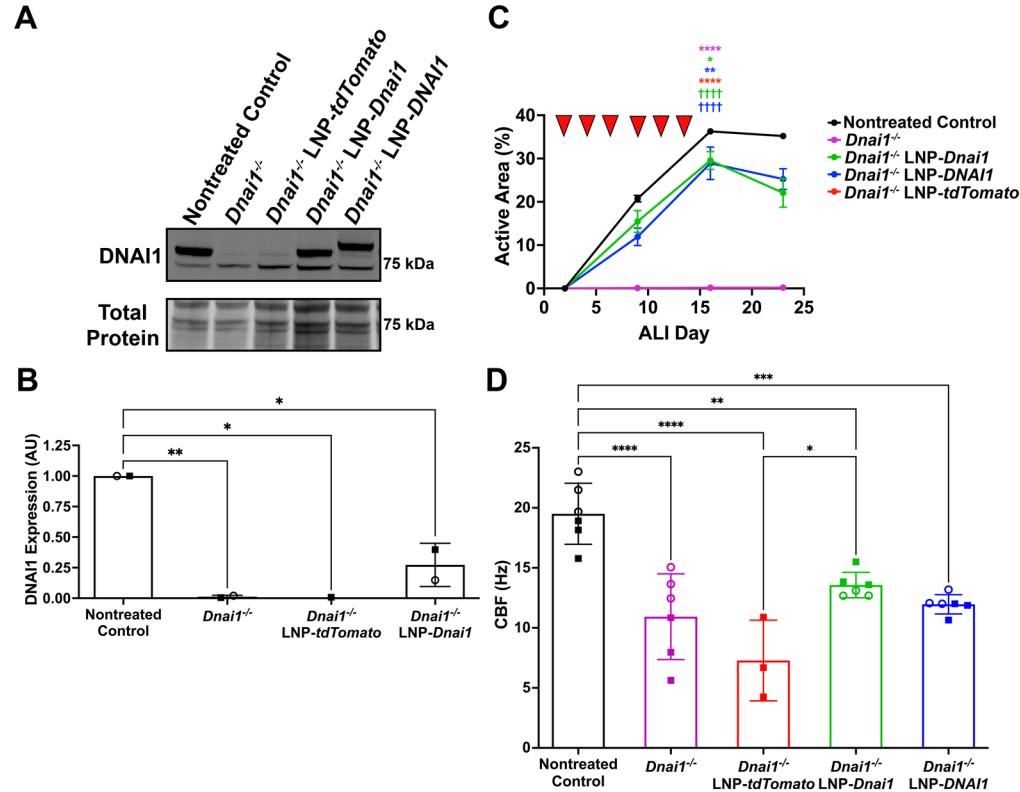

**Fig. 6. Treatment with human mRNA SORT-LNP-*DNAI1* rescues ciliary activity in *Dnai1*$^{-/-}$ mNPEC cultures.** Differentiating *Dnai1*$^{-/-}$ murine nasopharyngeal epithelial cell (mNPEC) cultures were treated basolaterally three times per week with 5 µg/ml SORT-LNP on ALI days 2, 4, 6, 9, 11 and 13 (red arrowheads in C) for a total of six times. (A) Representative immunoblot probing for DNAI1 on whole cell lysates of *Dnai1*$^{flox/flox}$ and *Dnai1*$^{-/-}$ mNPEC cultures. (B) Quantification of wild-type (80 kDa) DNAI1 protein levels in immunoblots represented in A. Raw signals were normalized to total protein stain, and then control ratios were set to 1.0. Data are mean±s.d. from two independent experiments. The levels of DNAI1 protein in *Dnai1*$^{-/-}$ mNPEC cultures treated with SORT-LNP-*DNAI1* were not quantified due to not being able to compare exogenous DNAI1 protein levels to endogenous DNAI1 protein levels in the mouse. (C) Active area measurements taken at ALI days 2, 9, 16 and 23 are shown as mean±s.d. This study was performed for two independent experiments in triplicate, with the replicate data presented in Fig. S5. Based on a one-way ANOVA combined with a Tukey's multiple comparisons test, the mean active areas on ALI day 16 of the treated mNPEC cultures were significantly different when compared to the nontreated control cultures (*$P$<0.05; **$P$<0.01; ****$P$<0.0001) and when compared to the *Dnai1*$^{-/-}$ cultures (††††$P$<0.0001). (D) Ciliary beat frequency (CBF) measurements on ALI day 23 or 27. Data shown are mean±s.d. from two independent experiments performed in triplicate. Open circle and square datapoints represent mean CBF from individual cultures from the first and second replicate experiment, respectively. For B and D, a one-way ordinary ANOVA followed by Tukey's multiple comparison test determined a significant difference between the means of nontreated and treated mNPEC cultures (*$P$<0.05; **$P$<0.01; ***$P$<0.001; ****$P$<0.0001).

with a long cellular half-life, which may prolong the restoration of mucociliary clearance (MCC). The persistent activity of the rescued motile cilia is likely also due to the slow rate of turnover of multiciliated cells in the mouse airway epithelium, where terminally differentiated multiciliated cells have been shown to have an average half-life of 6 months in the mouse trachea and 17 months in the lungs (Rawlins and Hogan, 2008; Roberson et al., 2020). Similarly, the human airway epithelium is thought to fully turn over every 1–4 months, which provides the possibility of less frequent dosing regimens for clinical applications of SORT-LNP-mRNA therapies (Bowden, 1983; Crystal et al., 2008; Tilley et al., 2015). To restore productive MCC in the conducting airways, repeated administrations of mRNA therapy will likely be required to efficiently penetrate mucus barriers and transfect a sufficient number of ciliated cells throughout the airways. Thus, the production of a stable DNAI1 protein with a long half-life is expected to enable greater accumulation of rescued activity over repeated administrations, especially considering the long half-life reported for airway ciliated cells (Bowden, 1983; Crystal et al., 2008; Rawlins and Hogan, 2008; Roberson et al., 2020; Tilley et al., 2015).

We observed that differentiating cultures with cells undergoing ciliogenesis and fully differentiated cultures both show rescued ciliary activity after treatment with SORT-LNP-encapsulated *Dnai1* mRNA, suggesting there may be multiple mechanisms to rescue ciliary activity. While undergoing ciliogenesis, differentiating multiciliated precursor cells can incorporate the newly made DNAI1 protein into their developing ciliary axonemes (Braschi et al., 2022; Fowkes and Mitchell, 1998; Lyu et al., 2024). However, ciliary rescue in fully differentiated cultures could occur by additional mechanisms that may not be mutually exclusive. One possibility is that in terminally differentiated ciliated cells, functional ODAs could be incorporated into the ciliary axoneme through the turnover or exchange of dynein arms during cilia maintenance (Dai et al., 2018). Another possibility is that functional multiciliated cells replace non-functional cells during cell turnover in these cultures. In this mechanism, *Dnai1* mRNA is transfected into ciliated precursor cells (i.e. basal or club cells) and due to the long half-life of the DNAI1 protein, subsequent differentiation into a functional multiciliated cell occurs. While it has been reported that less than 1% of the total airway cell population is actively proliferating in a healthy human lung (Boers et al., 1998), the

turnover rate is unknown in individuals with PCD. Rescuing mature cilia would likely lead to a more rapid response to mRNA therapy, rather than depending solely on the rescue that occurs during ciliogenesis and the natural turnover of the respiratory epithelium. In our studies, we observed modest but significant ($P<0.0001$, unpaired $t$-test) increases in active area 72 h after a single treatment of SORT-LNP-*Dnai1* mRNA (Fig. 5C). Since it takes ∼10 days for a cilium to be fully formed in mouse trachea after initiation of differentiation (You et al., 2002), we see evidence for the mechanism whereby fully differentiated cells turnover their dynein arms or exchange axonemal components in mature cilia during cilia maintenance. However, further studies are necessary to confirm the restoration and proper distribution of functional ODAs throughout the rescued axoneme. Therefore, understanding both the duration and mechanisms behind rescued motile ciliary activity could help estimate what duration of treatment, associated with various doses and dosing regimens, will be required to generate a long-term beneficial effect in individuals with PCD.

Multiple aspects of this *DNAI1* mRNA therapy need to be considered to determine if a beneficial effect in PCD-affected individuals will be observed, including the amount of rescued ciliary activity required for restoration of coordinated MCC and the tolerability of the dosing regimen to reach this required amount. Previous studies have suggested that a range of at least 20–30% of ciliary activity is sufficient to maintain MCC that results in no or a very mild PCD phenotype. For example, treatment of *Dnai1*$^{flox/flox/}$CreER$^{+}$ mice with different doses of tamoxifen showed that the rate of MCC was correlated with the level of intact genomic *Dnai1*, where animals with at least 20% of normal *Dnai1* levels exhibited MCC with no mucus accumulation and no sinusitis phenotype (Ostrowski et al., 2014). Additionally, PCD-affected female individuals with variants in the X-linked PCD gene *DNAAF6* have been shown to have variable levels of normal multiciliated cells due to random or skewed X-chromosome inactivation. Using *in vitro* and *in vivo* ciliary clearance assays of these individuals, a range of 30–62% of functioning multiciliated cells was shown to be sufficient to produce normal or slightly abnormal MCC that resulted in no or only mild respiratory symptoms (Loges et al., 2024). Within 1 week of treating fully differentiated *Dnai1*$^{-/-}$ mNPEC cultures with SORT-LNP-*Dnai1* mRNA three times a week, cultures were able to reach 32% of normal ciliary activity levels that eventually increased to a maximum of 65% of normal ciliary activity by ALI Day 38 (Fig. 5), suggesting that treatment of SORT-LNP-*DNAI1* mRNA can rescue the required amount of ciliary activity to restore productive MCC. However, further evaluation is required to determine if treatment through apical delivery and *in vivo* can rescue this level of coordinated ciliary activity to produce effective MCC, where some promising results have been shown with aerosolized delivery of SORT-LNP-*DNAI1* mRNA to human bronchial epithelial cells and nonhuman primates (Hennig et al., 2025). With the repeated administrations of SORT-LNP that are required with mRNA therapy to restore productive MCC, tolerability of the SORT-LNP formulations is another essential attribute of this potential therapeutic. While mNPEC cultures dosed at higher doses (10 or 20 μg/ml) showed signs of loss of cell culture integrity (culture leakiness, cell death and gaps in the epithelial layer), cultures dosed at lower doses (5 μg/ml) were able to reach significant levels of ciliary rescue without loss of cell culture integrity (Fig. 4). Additionally, Hennig et al. (2025) assessed the effects of transfection of SORT-LNP-mRNA on airway epithelium through aerosolized treatment on human bronchial epithelial cells by measuring cell distribution and LDH release as a readout of cytotoxicity. There were minimal changes between the cell type distribution and minimal LDH release in cultures treated with SORT-LNPs compared to nontreated controls. Additionally, nonhuman primates that were administered a single dose of SORT-LNP-*DNAI1* mRNA by inhaled delivery had no adverse reactions and showed no significant change in body temperature or mass (Hennig et al., 2025), suggesting that treatment with SORT-LNP-mRNA is tolerable at appropriate doses.

The results reported here add to the growing body of work to support an mRNA therapy as a treatment for PCD, specifically using LNPs to deliver mRNA to airway cells with immotile cilia and restore their ciliary activity. This work demonstrates that treatment of *Dnai1*$^{-/-}$ mNPEC cultures with SORT-LNPs containing either mouse *Dnai1* or human *DNAI1* mRNAs results in the expression of wild-type DNAI1 protein that is incorporated into ciliary axonemes, leading to substantial rescue that approaches normal levels of ciliary activity. Supported in part by these results, ReCode Therapeutics conducted a Phase 1 double-blind, placebo-controlled single ascending dose study (NCT05737485) in healthy subjects and an open-label single ascending dose study in individuals with PCD caused by *DNAI1* mutations to evaluate the safety and tolerability of inhaled RCT1100, a *DNAI1* mRNA-based therapeutic delivered by SORT-LNPs. In conclusion, SORT-LNP encapsulated mRNA is a promising potential treatment for primary ciliary dyskinesia that may overcome obstacles previously seen with other therapies and delivery modalities, and warrants further study, including expansion to the use of mRNAs for additional genes with known pathogenic variants that cause PCD.

## MATERIALS AND METHODS
### mRNA production
DNA plasmids containing polyA-tailed and *tdTomato* or sequence-optimized (HA-tagged and untagged) *Dnai1* and *DNAI1* were enzymatically linearized. All mRNA constructs were generated using DNA-template directed *in vitro* transcription using T7-RNA polymerase with N1-methyl-pseudouridine (m$^1$ψ) triphosphate substitution for naturally occurring uridine triphosphate. *In vitro* transcriptions were performed following standard conditions. Post-transcriptional enzymatic 5′-capping was performed as described by the manufacturer. 5′-capped mRNA samples were purified and resuspended in water. Subsequently, mRNA samples were passed through 0.22 μm filter and stored frozen at −80°C until they were formulated.

### Lipid nanoparticle formulation
*Dnai1*, *DNAI1* or *tdTomato* mRNA-loaded Selective Organ Targeting (SORT)-LNP were prepared via nanoprecipitation by mixing an aqueous (sodium citrate buffer, pH 4.0) and an organic (ethanol) solvent stream containing an ionizable lipid, a cationic SORT lipid (1,2-dimyristoyl-sn-glycero-3-ethylphosphocholine), 1,2-dioleoyl-sn-glycero-3-phosphoethanolamine (DOPE), cholesterol and PEG2k-DMG in a volumetric flow ratio of 3:1. Formulations were incubated at room temperature for 30 min prior to buffer exchange and concentration, and subsequently passed through 0.22 μm filter. SORT-LNP formulations were prepared in a single batch and aliquoted into smaller volumes before long-term storage at −80°C. The required number of aliquots were thawed at room temperature before treatments and leftover aliquots were stored at 4°C for up to 7 days and then discarded. Stability data of untagged SORT-LNP-*DNAI1* mRNA formulations demonstrated minimal change in particle characteristics and potency for up to 24 months at −80°C and for up to 1 week at 4°C. All experiments used a single batch for all LNP formulations, except the untagged and HA-tagged mouse *Dnai1* mRNA formulations, where multiple batches were used. All LNP formulations were determined to have a particle size in the range of 62–118, polydispersity index of 0.1, mRNA encapsulation efficiency in the range of 48.1–98.2, and mRNA payload concentration in the range of 0.50–1.08 mg/ml (Table S1).

## Conditional *Dnai1* knockout mouse model

All animal protocols and procedures were reviewed and approved by the University of North Carolina at Chapel Hill (UNC-CH) Institutional Animal Care and Use Committee (IACUC). The generation, breeding and maintenance of *Dnai1*$^{flox/flox}$ mice has previously been described (Ostrowski et al., 2010). In brief, transgenic mice carrying loxP sites flanking exons 17 and 18 of the mouse *Dnai1* gene (*Dnai1*$^{flox/flox}$) were maintained on a mixed background (C57Bl6/129). These animals were bred with mice expressing a Cre recombinase under the control of a Rosa promoter that is inducible with tamoxifen treatment (R26CreER; Jackson Laboratory). Experimental animals were generated by mating *Dnai1*$^{flox/flox}$/CreER$^{-/-}$ mice with *Dnai1*$^{flox/flox}$/CreER$^{+/-}$ mice.

## Mouse nasopharyngeal epithelial cell cultures

mNPEC cultures were derived from the isolation and digestion of the nasopharynx tissue from *Dnai1*$^{flox/flox}$/Cre$^{+/-}$ male and female mice of 6 months or less, followed by the expansion in conditional reprogramming cell (CRC) conditions and plating of the murine nasopharyngeal epithelial cells at an air/liquid interface (ALI) using modifications of previously described methods (Antunes et al., 2007; Eenjes et al., 2018; Ostrowski et al., 2010; Suprynowicz et al., 2012; Woodworth et al., 2007; You et al., 2002). Mice were euthanized by carbon dioxide ($CO_2$) inhalation and aortic exsanguination, followed by removal of the lower jaw. Tissue was resected from the posterior opening of the pharyngeal duct to the posterior hard palate. After removal of connective tissue, the nasopharynx was collected on ice in F12 nutrient mix with 1×penicillin-streptomycin and then washed for 5 min at room temperature on an orbital shaker. Nasopharyngeal tissues were digested for 1.5 h at room temperature on an orbital shaker in accutase supplemented with 0.5 mg/ml pronase. Following digestion, the accutase enzyme solution was neutralized, the remaining tissues were removed from the mixtures and the cell suspensions were centrifuged for 2 min at 1000 *g* at 4°C. Cells were resuspended, plated and expanded in CRC conditions on PureCol-coated culture dishes with mitomycin-treated 3T3-J2 fibroblast feeder cells in CRC media supplemented with 10 μM Y-27632 (Enzo Life Sciences), a Rho-associated kinase (ROCK) inhibitor. Mitomycin-treated feeder cells were previously created by treating fibroblasts with 4 μg/ml Mitomycin C for 3 h at 37°C, and were plated on PureCol-coated culture dishes 24 h before plating of mNPEC cultures. Base CRC medium contains a 3:1 volume ratio of Dulbecco's modified Eagle's medium (DMEM) with 4.5 g/l D-glucose, L-glutamine, and 110 mg/l sodium pyruvate and F12 nutrient mix, supplemented with 1×penicillin-streptomycin, 25 ng/ml hydrocortisone, 25 ng/ml epidermal growth factor, 5 μg/ml insulin, 250 ng/ml amphotericin B, 10 μg/ml gentamicin, 1 nM cholera toxin and 7.5% fetal bovine serum (FBS).

Once cultures became confluent, mNPEC cultures were passaged (p1) and expanded again in CRC conditions. A subset of cells were treated with 1 μM tamoxifen at the time of plating and with each media change for a total of three to four times or 6 to 8 days of tamoxifen treatment during the second expansion phase (*Dnai1*$^{-/-}$ cells). Once cultures became confluent, mNPEC cultures were passaged (p2) and seeded onto collagen-coated Millicell inserts (Millipore, 12 mm, hydrophilic PTFE membrane, 0.4 μm pore size) at 1.2×10$^5$ cells per insert (0.6 cm$^2$ surface area). Additional cells were cryopreserved for future thawing and experiments. After plated cultures reached confluence, apical medium was removed to differentiate cultures at an air-liquid interface (ALI), as previously described (Antunes et al., 2007; You et al., 2002), and the basolateral media was changed to a 1:1 ratio of DMEM: F12 nutrient mix with 1×penicillin-streptomycin, 0.03% sodium biocarbonate, 10% KnockOut Serum Replacement, 50 nM retinoic acid and 2% MDM media. MDM media is composed of 1:1 ratio of DMEM: F12 nutrient mix with 1×penicillin-streptomycin, 0.03% sodium bicarbonate, 0.1% bovine serum albumin, 1×insulin-transferrin-selenium, 25 ng/ml cholera toxin, 5 ng/ml murine epidermal growth factor, 0.03 mg/ml bovine pituitary extract and 50 nM retinoic acid. mNPEC cultures were washed for 5 min at 37°C with PBS twice per week to prevent mucus accumulation, and medium was replaced with fresh, basolateral medium three times a week. mNPEC cultures were considered fully differentiated at culture day 20, as this was usually when ciliary active area would reach its maximum and start to plateau. Basolateral treatments of cultures with SORT-LNP-mRNA were performed starting at ALI day 2–4 for differentiating cultures or ALI day 17–20 for well-differentiated cultures. Cultures were treated with various concentrations of SORT-LNP-mRNA in 1 ml of basolateral medium and incubated at 37°C for 4 h three times per week for up to eight treatments in total. The specific treatment regimen for each experiment is detailed in the main text and figure legends.

## Ciliary active area and ciliary beat frequency measurements

Ciliary active area and CBF measurements were performed as previously described (Sears et al., 2015; Sisson et al., 2003). Briefly, a maximum of three cultures from each experimental group were apically washed with 1×PBS, pH 7.4 for 5 min at 37°C, followed by an additional 1×PBS apical wash at room temperature. 35 μl of 1×PBS was added to each culture apically, and ciliary activity was visualized with a Nikon Eclipse TE2000 inverted microscope (Nikon Instruments) using a 20×/0.45 NA objective and phase optics. Temperature was maintained at 37°C and $CO_2$ levels were maintained at 5% using an OkoLab incubator. Using a Basler acA1300-200 μm camera, high-speed videos (60 fr/s) were recorded and analyzed using Sisson Ammons Video Analysis (SAVA) software (Ammons Engineering, Clio, MI) to determine the ciliary active area and CBF via whole-field analysis (Sisson et al., 2003). A total of 12–14 alternating fields were recorded, analyzed and averaged from each culture measured.

## High-resolution ciliary activity videos

Cells were scraped from Millicell culture membranes by breaking off the culture feet, separating the membrane from the culture side walls and drawing a coverglass across the flattened membrane. The cells in media were then transferred to a glass-bottom dish (27 mm, Thermo Scientific, 150682). A coverglass (No.1, 18 mm diameter) held up with silicone grease was placed on top leaving two ends open to the air. The assembly was then placed in an OkoLab incubator on a Nikon Eclipse TE-2000 inverted microscope. The sample chamber was set to 37°C, 5% CO2, humidified air and a 400 ml/min flow rate. Cells were imaged using a 60× oil objective (NA=1.4) with DIC-H optics and 2× post-objective magnification. Videos were captured at 300 fr/s using a Basler acA1300-200um camera controlled by SAVA software (Ammons Engineering, Clio, MI, USA). Movies were created by enhancing the contrast and using a replay speed of 30 fr/s (0.1× live).

## Whole-mount immunofluorescent staining

Whole-mount immunofluorescent staining was performed as previously described (Smith et al., 2022). mNPEC cultures were washed apically with 1×PBS, pH 7.4 at 37°C for 5 min, followed by additional apical and basolateral washes with 1×PBS at room temperature. Cultures were fixed with 4% paraformaldehyde in 1×PBS for 30 min at room temperature and then washed three times with 1×PBS for 5 min at room temperature. After separation from the Millicell insert using microscissors, membranes were cut into two pieces and processed in 24-well plates. Membranes were permeabilized with 0.2% Triton-X in 1×TBS for 30 min at room temperature, followed by two 1×PBS washes for 5 min at room temperature. To prevent nonspecific antibody binding, membranes were blocked in 1% bovine serum albumin, 1% fish gelatin, 0.1% Triton-X and 5% donkey serum in 1×TBS for 1 h at room temperature. Membranes were then incubated with either primary antibodies or isotype-matched control antibodies in blocking solution overnight at 4°C. Following three washes with 0.25% bovine serum albumin, 0.25% fish gelatin, 0.25% Triton-X and 1.25% donkey serum in 1×TBS for 20 min at room temperature, membranes were incubated with secondary antibodies in blocking solution for 1 h at room temperature protected from light. Membranes were then washed with 1×TBS for 15 min at room temperature four times, followed by mounting on slides using ProLong Diamond Antifade Reagent (Thermo Fisher). Nuclei were stained using DNA dye Hoechst33342 (Thermo Fisher).

Slides were imaged using a Zeiss 800 upright confocal microscope with a 40×/1.3 oil objective with Zeiss Zen Blue software. A signal was not observed for staining of mNPECs using isotype-matched control antibodies. FIJI software and/or Adobe Photoshop was used to pseudo-color and adjust the brightness/contrast evenly across samples per experiment for publication. For quantification of transfection efficiency, FIJI software via

the Analyze Particles function was used to count the total number of cells and total number of fluorescently positive cells after thresholding and watershed separation. All antibodies and their dilutions used are listed in Table S2.

## Immunoblots

Immunoblots probing for DNAI1 and HA-tag were performed as described previously (Smith et al., 2022). mNPEC cultures were washed with 1×PBS, pH 7.4 for 5 min at 37°C twice and then were lysed in RIPA buffer (Thermo Fisher Scientific) with 1× protease inhibitor cocktail (Sigma-Aldrich). The protein concentration of whole-cell lysates was calculated using a Pierce BCA Protein Assay kit (Thermo Fisher Scientific). 10–20 µg of protein lysates were loaded and electrophoresed on Novex NuPAGE 4-12% Bis-Tris gels in 1×NuPAGE MOPS-SDS Running Buffer (Thermo Fisher) for 47 min at 200 V. Gels were then transferred to 0.45-µm nitrocellulose membranes using 1×NuPAGE Transfer Buffer with 10% methanol. Membranes were then stained with REVERT Total Protein Stain kit (LI-COR) after transfer, imaged using a LI-COR Odyssey Scanner and then destained. Membranes were blocked in 5% non-fat milk in 1×TBS for 1 h at room temperature, followed by incubation in primary antibodies diluted in 5% non-fat milk in 1×TBST (0.1% Tween 20) at 4°C overnight. Following four washes with 1×TBST (0.1% Tween 20) for 5 min at room temperature, membranes were incubated in secondary antibodies diluted in 5% non-fat milk in 1×TBST (0.1% Tween 20). Membranes were washed four times with 1×TBST (0.1% Tween 20) for 5 min at room temperature, and then were scanned using a LI-COR Odyssey Scanner. All primary and secondary antibodies and their dilutions are listed in Table S2. Full immunoblots are provided for immunoblot data transparency in Fig. S6.

Fluorescent immunoblots were quantitatively analyzed using ImageStudio Lite software (LI-COR). Background-subtracted signals were normalized to a total protein stain, and then control means were then set to 1.0. The brightness/contrast of immunoblot images was adjusted for publication with FIJI software.

## Statistics

Data from all experiments are presented as the mean±s.d. All experiments were performed with at least two independent replicates. Prism 10 software (GraphPad) was used to perform the following statistical tests: one-way analysis of variance (ANOVA) with Tukey's multiple comparisons and unpaired two-tailed $t$-tests. To compare mean active areas between experimental groups, a one-way ANOVA was performed on mean active area measurements taken at the next SAVA measurement after the last dose of SORT-LNP-mRNA. The statistical method and $P$-value, where $P<0.05$ was considered significant, are listed in the figure legends for each experiment.

## Acknowledgements
The authors thank Dr Gang Chen and Dr Ling Sun in the Marsico Lung Institute at the University of North Carolina at Chapel Hill for their expertise in expansion and culture of mouse nasopharyngeal epithelial cells. We thank Maninder Sidhu for mRNA production, and Dr Lalithasri Ramasubramanian, Emmanuel Fasusi and Alanna Manning for producing LNP-encapsulated mRNA formulations. We also thank Henry Gong for his assistance in performing preliminary studies testing previous LNP formulations. We thank the University of North Carolina at Chapel Hill's Hooker Imaging Core Facility for providing advice and expertise for the microscopy studies.

## Competing interests
A.J.S., M.H., R.B.B., D.I., T.N.L., D.J.L. and B.A.W. disclose financial interests in ReCode Therapeutics. The remaining authors declare no competing interests.

## Author contributions
Conceptualization: A.J.S., M.H., R.B.B., D.J.L., B.A.W., L.E.O.; Data curation: A.J.S., P.R.S., W.Y., H.G., D.I.; Formal analysis: A.J.S., P.R.S.; Funding acquisition: D.J.L., L.E.O.; Investigation: M.H., R.B.B., T.N.L., B.A.W., L.E.O.; Methodology: A.J.S., P.R.S.; Project administration: T.N.L., D.J.L., B.A.W., L.E.O.; Resources: M.H., R.B.B., D.I., T.N.L., D.J.L., B.A.W.; Supervision: M.H., R.B.B., T.N.L., B.A.W., L.E.O.; Visualization: A.J.S., P.R.S.; Writing – original draft: A.J.S.; Writing – review & editing: A.J.S., P.R.S., M.H., R.B.B., T.N.L., D.J.L., B.A.W., L.E.O.

## Funding
This research was supported by a Sponsored Research Agreement with ReCode Therapeutics. Open Access funding provided by the University of North Carolina at Chapel Hill. Deposited in PMC for immediate release.

## Data and resource availability
All relevant data and details of resources can be found within the article and its supplementary information.

## Peer review history
The peer review history is available online at https://journals.biologists.com/jcs/lookup/doi/10.1242/jcs.264068.reviewer-comments.pdf

## Special Issue
This article is part of the Special Issue 'Cilia and Flagella: from Basic Biology to Disease', guest edited by Pleasantine Mill and Lotte Pedersen. See related articles at https://journals.biologists.com/jcs/issue/138/20.

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
