## [Peer Review File · Journal of Cell Science]

Lipid nanoparticle-encapsulated *Dnai1* mRNA rescues ciliary activity in primary ciliary dyskinesia mouse cell models

Amanda J. Smith, Patrick R. Sears, Mirko Hennig, Rumpa B. Bhattacharjee, Weining Yin, Hannah Golliher, Daniella Ishimaru, T. Noelle Lombana, David J. Lockhart, Brandon A. Wustman and Lawrence E. Ostrowski
DOI: 10.1242/jcs.264068

Editor: Pleasantine Mill

Review timeline

Original submission:	9 April 2025
Editorial decision:	22 May 2025
First revision received:	25 August 2025
Accepted:	4 September 2025

Original submission

First decision letter

MS ID#: jcs.264068

MS TITLE: Lipid Nanoparticle-encapsulated Dnaic1 mRNA Rescues Ciliary Activity in Primary Ciliary Dyskinesia Mouse Cell Models

AUTHORS: Amanda J. Smith; Patrick R. Sears; Mirko Hennig; Rumpa B. Bhattacharjee; Weining Yin; Hannah Golliher; Daniella Ishimaru; T. Noelle Lombana; David J. Lockhart; Brandon A. Wustman; Lawrence E. Ostrowski

ARTICLE TYPE: Research Article

Dear Larry and team,

Thank you for your patience, we have now reached a decision on the above manuscript as part of our Special Issue on Cilia and Flagella for Journal of Cell Science.

To see the reviewers' reports and a copy of this decision letter, please go to:

As you will see, the reviewers gave favourable reports but raised some key points for additional information and/or discussion that will require amendments to your manuscript. I hope that you will be able to carry these out because I would like to be able to accept your paper, depending on further comments from reviewers.

My overview, this manuscript by Smith et al represents a well presented and executed study providing key preclinical data in mouse airway cells behind the recently published Hennig et al PNAS paper 'Inhaled DNAI1 mRNA therapy for treatment of primary ciliary dyskinesia' <https://doi.org/10.1073/pnas.2421915>. It clearly details important background data to this first in man gene therapy for more the motile ciliopathy PCD. Both reviewers were very positive in their praise of the quality and depth of the data, with both requesting more details and more discussion in this minor revision. From an editorial standpoint, I would recommend the authors to change the protein symbol nomenclature for mouse to DNAI1 throughout (see <https://www.informatics.jax.org/mgihome/nomen/gene.shtml#ps> - all caps and no italics) and also align with UNIPROT database. It will simplify labels and text throughout. For Figure 2C, as reviewer 2 mentions, it is important to have a larger higher magnification insert view of the HA-tagged DNAI1 rescue to show localization and levels. Consider showing the single channel as grey scale for the HA- the human eye is better at reading signal intensity without color on black and we get a deeper

dynamic range. The legend of 2C is also lacking what time point this represents. Finally here, do the authors have data on endogenous DNAI1 antibody staining and rescue in both tagged and non-tagged mRNA as would be 2C- lower level protein expression/stability from WB also at the level of IF? Minor points- missing scale bar in Fig 1A and legends.

Reviewer 1

SUMMARY OF THE ADVANCE MADE IN THIS PAPER AND ITS POTENTIAL SIGNIFICANCE TO THE FIELD

Primary Ciliary dyskinesia: Primary ciliary dyskinesia (PCD) is a recessively inherited respiratory disease caused by abnormal structure and function of motile cilia. Dysmotility of the cilia leads to impaired mucociliary clearance and progressive respiratory disease with recurrent respiratory tract congestion and infections, leading to destructive lung disease (bronchiectasis). Hearing defects can arise from otitis media while other symptoms may include infertility and laterality defects including heart defects. However, the respiratory defect is the major cause of morbidity and mortality. No long-term randomised drug trials for PCD have ever been performed despite its frequency and current treatments are only empirically directed at symptoms, e.g., antibiotic therapy. PCD can be caused by mutations in any one of more than 50 currently known genes that encode proteins involved in the structure, assembly and generation of cilia. Recent developments in gene therapy, mRNA therapy, CRISPR gene editing, splice correcting and ASOs, have led to the hope that all PCD patients may one day be treatable by targeting the genetic basis of their disease.

Although the prospect of correcting or replacing expression of 50 different genes offers a challenge in developing gene therapies for PCD, the most prevalent pathogenic variants affect just a few genes including the dyneins DNAH5, DNAH11, DNAI1 and coiled-coil nexin link protein genes CCDC39 and CCDC40. LNP mRNA therapies are of interest for genetic therapies following on from their safety and efficacy in the SARS-CoV2 vaccines. This paper focuses on the SORT LNPs delivering Dnaic1 mRNA treatment targeted at PCD associated with deficiency of DNAI1, or Dnaic1 in mice, which leads to loss of outer dynein arms and ciliary dysmotility. The cellular model in which the therapy is tested comprises airway epithelial cells from a conditional Dnaic1 $-/-$ mice generated in the author's lab.

Results have shown that the LNPs efficiently transfect murine ALI cultures with tdTomato mRNA by basolateral delivery. Transfections of cells with DNAic1 mRNA or a HA tagged version, led to detection of DNAic1 in ciliary axonemes by immunohistochemical analysis and by Western blot analysis. This led to widespread restoration of ciliary motility with the correct CBF. Interestingly restoration of CBF occurred after transfection either of preciliated cells or mature differentiated ALI cultures suggesting correction either during ciliogenesis or by rescue of immotile cilia by the ciliary maintenance mechanisms. DNAI1 mRNA was equally as effective in restoring ciliary motility as murine Dnaic1, reflecting their highly conserved sequences. Transfection conditions were optimised for dosage quantity and repeat dosing.

SUGGESTIONS TO AUTHORS

The findings in the paper are well justified and the paper is overall clear and well written. I have the following questions that it would help to clarify.

- * What were the properties of the SORT RNA LNPs with respect to size and charge and efficiency of packaging mRNA? For repeat dosing were they freshly prepared each time or prepared in a single batch. If the latter how were they stored between doses and what is the LNP stability for transfection and biophysical properties?
- * The method of transfection was basolateral but in the trial delivery is by nebulised airway delivery as I understand. Why was the basolateral route of delivery used instead of apical in the ALI cultures? Do these murine ALI cultures produce copious amounts of mucus?
- * How does the mRNA reach the ciliated cells in differentiated ALI cultures by basolateral delivery if the epithelium is intact? Were TEER measurements performed pre- and post-transfection to assess the effects of transfection on the epithelium?

* What is the amount of cellular/ ciliary correction required for restoration of coordinated mucociliary clearance in vivo.

Minor errors

last line p16 : PBS-/-

Reviewer 2

SUMMARY OF THE ADVANCE MADE IN THIS PAPER AND ITS POTENTIAL SIGNIFICANCE TO THE FIELD:

This manuscript by Smith et al. describes a method for successful delivery of a motile ciliary dynein mRNA (*Dnaic1*) to cultured cells from a mouse model with primary ciliary dyskinesia (PCD) due to a mutation in the *Dnaic1* gene. Given that therapies for PCD remain limited, this report of an mRNA therapy is an important advancement in the field. The manuscript is well-written, and the study is quite meticulous in its design and execution. The data are sound and well controlled, the statistical analyses are appropriate, and the methods are thoroughly described. This is a strong study, and it is just recommended that the authors address the following minor comments.

SUGGESTIONS TO AUTHORS:

1. FIGURE 2C: The zoomed in box showing *Dnaic1*-HA in ciliary axonemes is small and a little hard to see. Given the importance of this localization data, it is recommended that the image be enlarged.
2. RESULTS (page 10): The authors state that "CBF measurements on ALI day 38 or 40 demonstrated restoration of mean CBF to within a normal range when differentiated *Dnaic1*^{-/-}-mNPEC were treated with SORT-LNP-*Dnaic1* or SORTLNP-*Dnaic1*-HA." While the normal range has previously been defined, both results here show a statistically significant decrease from the nontreated control. The authors should clarify whether this significant difference is likely to be biologically relevant.

First revision

Author response to reviewers' comments

Dear Dr. Mill,

Thank you for your time and consideration of our manuscript entitled "Lipid Nanoparticle-encapsulated *Dnaic1* mRNA Rescues Ciliary Activity in Primary Ciliary Dyskinesia Mouse Cell Models" for publication in the Special Issue on Cilia and Flagella of *Journal of Cell Science*. We appreciate the opportunity to submit a revised draft of the manuscript and the thoughtful feedback and suggestions provided to improve our paper. We agree that the topics and concepts that were suggested deserved more discussion in the context of our work and have incorporated majority of the suggested changes in our manuscript. Please see below, in red, for a point-by-point response to the reviewers' and your comments and questions. All page numbers listed here refer to the revised manuscript, and all of the changes are also highlighted within the manuscript. Any references listed here are also included in the manuscript references list. We have also reformatted our manuscript, specifically the in-text citations and the references list, to align with *Journal of Cell Science*'s formatting guidelines and have also included any additional information, including a blot transparency supplemental figure and submission checklist, as requested.

Reviewers' Comments to the Authors:

Reviewer #1

The findings in the paper are well justified and the paper is overall clear and well written. I

have the following questions that it would help to clarify.

1. What were the properties of the SORT RNA LNPs with respect to size and charge and efficiency of packaging mRNA? For repeat dosing were they freshly prepared each time or prepared in a single batch. If the latter how were they stored between doses and what is the LNP stability for transfection and biophysical properties?

Thank you for suggesting to provide more information on the biophysical and stability properties of the SORT-LNP formulations and for clarification on storage of the SORT-LNPs. Below, we have included a table that lists the particle size, polydispersity index, zeta potential, encapsulation efficiency of the mRNA cargo, and the nominal mRNA concentration (mg/mL) of the LNP formulations. The LNP formulations were prepared in a single batch and were aliquoted in small volumes before long-term storage at -80°C . For the untagged or HA-tagged *DNAI1* mRNA and *tdTomato* mRNA, a single batch was used for all experiments; multiple batches were used of untagged or HA-tagged *Dnai1* mRNA. For repeat dosing, the required number of aliquots were thawed at room temperature before treatment, and any leftover aliquots were stored at 4°C for up to 7 days and then discarded. Stability data at -80°C and 4°C using untagged LNP-*DNAI1* mRNA formulations showed minimal change in particle characteristics and potency for up to 24 months at -80°C and for at least a week at 4°C .

In terms of changes to our manuscript, we have also included the formulation characteristics table as Table 1 in our Supplemental Data and adjusted the Antibodies Used table to Table 2 in our Supplemental Data. We have cited the new Supplemental Table 1 on page 6 of the Results section and page 17 in the Materials and Methods section of the manuscript by stating:

“All LNP formulations were determined to have a particle size in the range of 62 - 118 nm, polydispersity index of 0.1, mRNA encapsulation efficiency in the range of 48% - 98%, and mRNA concentration in the range of 0.5 - 1.08 mg/mL (Table S1).”

Supplemental Table 1. LNP Formulation Characteristics

Formulation	Particle Size (nm)*	Polydispersity Index	Zeta Potential (mV)	Encapsulation Efficiency (%)	Nominal mRNA Concentration (mg/mL)
LNP-formulated Mouse Dnai1 mRNA (ranges shown)	62 - 76	0.1	ND	94.0 – 98.2	0.77 – 1.08
LNP-formulated HA-tagged Mouse Dnai1 mRNA (ranges shown)	67 - 76	0.1	ND	92.0 – 98.1	0.86 – 0.95
LNP-formulated Human DNAI1 mRNA	70	0.1	ND	98.3	0.94
LNP-formulated HA-tagged Human DNAI1 mRNA	69	0.1	1.81	93.8	1.07
LNP-formulated tdTomato mRNA	118	0.1	-0.230	48.1	0.50

*Particle size was measure by Malvern dynamic light scattering unit; ND: not determined.

We have also added the following information in the LNP Formulations section of the Method and Materials on page 17:

“SORT-LNP formulations were prepared in a single batch and aliquoted into smaller

volumes before long-term storage at -80°C . The required number of aliquots were thawed at room temperature before treatments and leftover aliquots were stored at 4°C for up to 7 days and then discarded. Stability data of untagged SORT-LNP-*DNAI1* mRNA formulations demonstrated minimal change in particle characteristics and potency for up to 24 months at -80°C and for up to one week at 4°C . All experiments used a single batch for all LNP formulations except the untagged and HA-tagged mouse *Dnai1* mRNA formulations, where multiple batches were used.”

2. The method of transfection was basolateral but in the trial delivery is by nebulised airway delivery as I understand. Why was the basolateral route of delivery used instead of apical in the ALI cultures? Do these murine ALI cultures produce copious amounts of mucus?

Thank you for asking for the reasoning behind the method of delivery for the LNPs in our studies. While it is correct that the method of delivery for the LNPs is through nebulization in the clinical trial (NCT05737485), we delivered the LNPs to our mouse nasopharyngeal epithelial cell cultures (mNPEC) by basolateral treatment via dilution of the LNPs in the cell cultures’ basolateral media. While control and *Dnai1*^{-/-} cultures do produce mucus, the mucus accumulation in these cultures was not substantial to prevent apical transfection as we performed 5 min. PBS washes twice a week as part of the cell culture maintenance protocol. The reason why basolateral delivery was used for the LNPs was that the main purpose of the studies listed in this manuscript was to provide proof-of-concept data that uptake of the LNP-encapsulated *Dnai1* mRNA could lead to DNAI1 protein expression, localization into ciliary axonemes, and rescue of ciliary function; thus to maximize the transfection of the *Dnai1* mRNA, we treated mNPEC cultures basolaterally as it is known apical transfection of airway epithelial cell cultures at ALI has been previously difficult and limited due to the mucus layer and cellular barrier (Bañuls et al., 2020). In addition, we determined treating mNPEC with an apical liquid bolus of the LNPs for 5 hours repeatedly made the mNPEC cell cultures more prone to losing integrity of their cellular barrier, causing leakiness, creation of holes, and “crashing” of the cell culture. We attributed these effects due to re-submersion of cultures in liquid that are meant to be at an air-liquid interface (ALI), as treatment with the LNPs through basolateral delivery at an appropriate dose (5 $\mu\text{g}/\text{mL}$ or below) did not lead to these effects. We also wanted to prevent any changes in the mNPEC cultures’ normal characteristics due to repeated submersions of ALI cultures, where previous studies have shown re-submersion of primary rat tracheal epithelial cells and human bronchial epithelial cells can lead to inhibition of ciliated cell differentiation, impaired epithelial cellular barrier integrity, alteration of transcriptomes and cell signaling pathways, and increased secretion of pro-inflammatory cytokines and growth factors (Clark et al., 1995; Gerovac et al., 2014; Mallek et al., 2024). However, studies to explore clinically relevant delivery methods were assessed by our colleagues, where *DNAI1*-knockdown human bronchial epithelial cell cultures were treated with aerosolized SORT-LNPs encapsulating human *DNAI1* mRNA, which demonstrated successful transfection and translation of a functional DNAI1 protein that resulted in rescued ciliary activity without losing integrity of the cell cultures (Hennig et al., 2025).

To explain our reasoning further in the manuscript, we included the following statements on page 6 in the Results section:

“To maintain the air-liquid interface of the cultures, limit any effect of re-submersion of the cultures, including inhibition of ciliated cell differentiation, impairment of the cellular barrier integrity, and alteration of cell signaling pathways (Clark et al., 1995; Gerovac et al., 2014; Mallek et al., 2024), and maximize transfection of the LNPs, mNPEC were treated basolaterally with the SORT-LNP-mRNA formulations in all studies.”

We have also included this statement to give further information about our cell culture maintenance protocol for mucus accumulation in the Materials and Methods section for Mouse Nasopharyngeal Epithelial Cell Cultures on page 19:

“mNPEC cultures were washed for 5 min at 37°C with PBS twice a week to prevent mucus accumulation, and media was replaced with fresh, basolateral media three times a week.”

3. How does the mRNA reach the ciliated cells in differentiated ALI cultures by basolateral delivery if the epithelium is intact? Were TEER measurements performed pre- and post-transfection to assess the effects of transfection on the epithelium?

Thank you for asking for clarification about how the ciliated cells are transfected by

basolateral delivery and their effects on the epithelium. While ciliated cells are mostly exposed at the apical surface of the airway epithelium, all cells of the pseudostratified airway epithelium, including all ciliated cells, are in contact with the basement membrane of the epithelium and thus, can be transfected directly by basolateral delivery. Through basolateral delivery of LNP-tdTomato mRNA in mNPEC cultures, we detected transfection of approximately 40% of cells present in fields of view by positive tdTomato fluorescence signal (Figure 1B). Further evaluation by flow cytometry of human bronchial epithelial cells transfected with SORT-LNP-mRNA through basolateral delivery demonstrated that approximately 90% of cells transfected were basal cells and 10% of cells transfected were ciliated cells (Hennig et al., 2025). While we do not have TEER measurements of our mNPEC cultures either pre- or post-transfection of the LNP formulations, Hennig et al. assessed the effects of transfection of SORT-LNP-mRNA on the epithelium through aerosolized treatment on human bronchial epithelial cells by measuring cell distribution and LDH release as a readout of cytotoxicity. There were minimal changes between the cell type distribution of the cultures and minimal LDH release with LNP treatment compared to nontreated controls (Hennig et al., 2025).

To demonstrate this information in the manuscript, we included the following text in the Discussion section on page 12:

“Due to all cells being in contact with the basement membrane in the pseudostratified airway epithelium present in mNPEC, basolateral treatment gives the potential for all cells to be transfected directly. Approximately 40% of cells of mNPEC were determined to be transfected by SORT-LNP- *tdTomato* mRNA through basolateral delivery (Fig. 1B). Hennig et al. (2025) determined by flow cytometry of human bronchial epithelial cells transfected with SORT-LNP-mRNA through basolateral or nebulized apical delivery that the cell type distribution of transfected cells includes approximately 90% basal cells and 10% ciliated cells with basolateral delivery, but more ciliated and secretory cells with apical delivery, which suggests that the majority of cells transfected in mNPEC may have been basal cells. Further evaluation is required to confirm the uptake and cell biodistribution of SORT-LNPs in mNPEC and how it is influenced by the method of delivery.”

We have also included the following text in the Discussion section on page 15:

“With repeated administrations of SORT-LNP that are required with mRNA therapy to restore productive MCC, tolerability of the SORT-LNP formulations is another essential attribute of this potential therapeutic. While mNPEC dosed at higher doses (10 or 20 $\mu\text{g}/\text{mL}$) showed signs of loss of cell culture integrity (culture leakiness, cell death, and gaps in the epithelial layer), cultures dosed at lower doses (5 $\mu\text{g}/\text{mL}$) were able to reach significant levels of ciliary rescue without loss of cell culture integrity (Fig. 4). Additionally, Hennig et al. (2025) assessed the effects of transfection of SORT-LNP-mRNA on airway epithelium through aerosolized treatment on human bronchial epithelial cells by measuring cell distribution and LDH release as a readout of cytotoxicity. There were minimal changes between the cell type distribution and minimal LDH release in cultures treated with SORT-LNPs compared to nontreated controls. Additionally, nonhuman primates that were administered a single dose of SORT-LNP-*DNAI1* mRNA by inhaled delivery had no adverse reactions and showed no significant change in body temperature or weight (Hennig et al., 2025), suggesting that treatment with SORT-LNP-mRNA is tolerable at appropriate doses.”

4. What is the amount of cellular/ ciliary correction required for restoration of coordinated mucociliary clearance in vivo.

We agree that this question deserves further discussion in our manuscript. Due to PCD mainly having recessive genetic inheritance, we know that at least 50% of normal protein levels, and potentially 50% of normal ciliary activity levels, is required to have coordinated mucociliary clearance with no clinical phenotype. Previous studies have suggested that a range of at least 20-30% of ciliary activity is sufficient to maintain MCC that results in no or a very mild PCD phenotype. For example, treatment of *Dnai1^{flox/flox}/CreER+* mice with different doses of tamoxifen showed that the rate of mucociliary clearance was correlated with the level of intact genomic *Dnai1*, where animals with at least 20% of normal *Dnai1* levels exhibited mucociliary clearance with no mucus accumulation and no sinusitis phenotype (Ostrowski et al., 2014). Additionally, PCD-affected female individuals with variants in the X-linked PCD gene, *DNAAF6*, have been shown to have variable levels of normal multiciliated cells due to random or skewed X-chromosome inactivation. Using *in vitro* and *in vivo* ciliary clearance assays of these individuals, a range of 30-62% of functioning

multiciliated cells was shown to be sufficient to produce normal or slightly abnormal MCC that resulted in no or only mild respiratory symptoms (Loges et al., 2024).

We have included the following text in our manuscript in the Discussion section on page 14-15: "Multiple aspects of this *DNAI1* mRNA therapy need to be considered to determine if a beneficial effect in PCD-affected individuals will be observed, including the amount of rescued ciliary activity required for restoration of coordinated MCC and the tolerability of the dosing regimen to reach this required amount. Previous studies have suggested that a range of at least 20-30% of ciliary activity is sufficient to maintain MCC that results in no or a very mild PCD phenotype. For example, treatment of *DNAI1*^{flox/flox}/CreER+ mice with different doses of tamoxifen showed that the rate of mucociliary clearance was correlated with the level of intact genomic *Dnai1*, where animals with at least 20% of normal *Dnai1* levels exhibited mucociliary clearance with no mucus accumulation and no sinusitis phenotype (Ostrowski et al., 2014). Additionally, PCD-affected female individuals with variants in the X-linked PCD gene, *DNAAF6*, have been shown to have variable levels of normal multiciliated cells due to random or skewed X-chromosome inactivation. Using *in vitro* and *in vivo* ciliary clearance assays of these individuals, a range of 30-62% of functioning multiciliated cells was shown to be sufficient to produce normal or slightly abnormal MCC that resulted in no or only mild respiratory symptoms (Loges et al., 2024). Within a week of treating fully differentiated *Dnai1*^{-/-} mNPEC with SORT-LNP-*Dnai1* mRNA three times a week, cultures were able to reach 32% of normal ciliary activity levels that eventually increased to a maximum of 65% of normal ciliary activity by ALI Day 38 (Fig. 5), suggesting that treatment of SORT-LNP-*Dnai1* mRNA can rescue the required amount of ciliary activity to restore productive MCC. However, further evaluation is required to determine if treatment through apical delivery and *in vivo* can rescue this level of coordinated ciliary activity to produce effective MCC, where some promising results have been shown with aerosolized delivery of SORT-LNP-*DNAI1* mRNA to human bronchial epithelial cells and nonhuman primates (Hennig et al., 2025)."

Minor errors

5. last line p16 : PBS-/-

Thank you for pointing this out. We have changed this text to simply read "PBS" on page 19.

Reviewer #2:

1. FIGURE 2C: The zoomed in box showing Dnaic1-HA in ciliary axonemes is small and a little hard to see. Given the importance of this localization data, it is recommended that the image be enlarged.

Thank you for this recommendation. We have increased the size of the zoomed in box in Figure 2C, as well as presented the single channels in grey scale for better visualization in both Figure 1A and 2C.

2. RESULTS (page 10): The authors state that "CBF measurements on ALI day 38 or 40 demonstrated restoration of mean CBF to within a normal range when differentiated *Dnaic1*^{-/-} mNPEC were treated with SORT-LNP-*Dnaic1* or SORTLNP-*Dnaic1*-HA." While the normal range has previously been defined, both results here show a statistically significant decrease from the nontreated control. The authors should clarify whether this significant difference is likely to be biologically relevant.

Thank you for your suggestion. In one of our studies, the CBF measurements of SORT-LNP-*Dnai1* treated *Dnai1*^{-/-} mNPEC were able to return to levels similar to *Dnai1*^{flox/flox} mNPEC with no significant difference between the two experimental groups (Figure 3B). However, we do acknowledge, in our other studies, the levels of CBF of SORT-LNP-*Dnai1* treated *Dnai1*^{-/-} mNPEC were able to return to a normal range, but the CBF levels was still significantly different than the nontreated controls. While it is possible that multiple different factors, including handling of cultures, temperature, hydration status and amount of mucus present on the cultures, could have influenced the CBF readings of these experiments (Sears et al., 2015), we also cannot rule out if the amount of exogenous *DNAI1* incorporated into the ciliary axonemes in the SORT-LNP-*Dnai1*

treated *Dnai1*^{-/-} mNPEC was not the same as the amount of endogenous DNAI1 incorporated in control mNPEC. Thus, there could be a sufficient amount of DNAI1 in ciliary axonemes to rescue motility with CBF on the lower range of normal, but perhaps not to the levels of DNAI1 protein incorporated in ciliary axonemes in *Dnai1*^{flox/flox} mNPEC to return to non-significantly different CBF levels. The amount and localization of rescued DNAI1 incorporation in ciliary axonemes could also influence the frequency and waveform of the rescued ciliary beat. Further evaluation is needed to determine the levels and localization of rescued DNAI1 in ciliary axonemes compared to controls, how they correlate to levels of rescued CBF, and if any difference in these levels would be clinically relevant.

We have changed this statement in our Results section on page 10-11 to: “CBF measurements on ALI day 38 or 40, while significantly decreased from nontreated controls, demonstrated restoration of mean CBF to within a normal range when differentiated *Dnai1*^{-/-} mNPEC were treated with SORT-LNP-*Dnai1* or SORT-LNP-*Dnai1*-HA mRNA (Rogers et al., 2022) (Fig. 5D).”

We have also included the following for further discussion of these results in our Discussion section on page 13:

“The rescued motile cilia had a mean CBF in the normal range of 8 to 18 Hz but did not return to *Dnai1*^{flox/flox} CBF levels in some cases (Fig. 5D, 6D) (Rogers et al., 2022). The lower CBF levels in treated *Dnai1*^{-/-} mNPEC could indicate that there was sufficient DNAI1 protein incorporated into the cilia to provide the force for motility to reach the lower range of normal CBF levels, but not to the levels required for rescuing CBF completely to *Dnai1*^{flox/flox} levels. Further analysis is required to correlate the amount of incorporated DNAI1 protein in rescued ciliary axonemes to the level of CBF achieved, if this has an impact on ciliary waveform and mucociliary transport, and if it is clinically meaningful.”

Editor’s Comments to the Authors:

1. From an editorial standpoint, I would recommend the authors to change the protein symbol nomenclature for mouse to DNAI1 throughout (see <https://www.informatics.jax.org/mgihome/nomen/gene.shtml#ps>- all caps and no italics) and also align with UNIPROT database. It will simplify labels and text throughout.

Thank you for this suggesting this. We have changed the nomenclature for mouse protein to DNAI1, as well as changed the gene and mRNA nomenclature to *Dnai1* throughout the manuscript’s text and in figure labels. We have also changed the text on page 4 from “*Dnai1* is the mouse homologue of human DNAI1, an ODA chain that is present along the length of the ciliary axoneme and is mutated in up to 10% of PCD cases (24-25).” to “DNAI1 is an outer dynein arm (ODA) chain that is present along the length of the ciliary axoneme and is mutated in up to 10% of PCD cases (Ostrowski et al., 2010; Zariwala et al., 2006).”

2. For Figure 2C, as reviewer 2 mentions, it is important to have a larger higher magnification insert view of the HA-tagged DNAI1 rescue to show localization and levels. Consider showing the single channel as grey scale for the HA- the human eye is better at reading signal intensity without color on black and we get a deeper dynamic range. The legend of 2C is also lacking what time point this represents.

Thank you for this recommendation. We have increased the size of the zoomed in box in Figure 2C, as well as presented the single channels in grey scale in Figures 1A and 2C for better visualization. We have also included in the legend for Figure 2C the text “mNPEC were fixed at ALI Day 20, approximately 10 days after the last treatment with SORT-LNP-mRNA.” on page 29.

3. Finally here, do the authors have data on endogenous DNAI1 antibody staining and rescue in both tagged and non-tagged mRNA as would be 2C- lower level protein expression/stability from WB also at the level of IF?

While we did perform staining for endogenous DNAI1 in our studies using a mouse anti-DNAI1 antibody, we observed a high level of background signal present in our fixed mNPEC cultures

that interfered with comparisons between experimental groups. The high level of background was suspected to be either due to non-specific staining due to the host species of the antibody or the presence of a high level of truncated DNAI1 in the cells as demonstrated in immunoblots of protein lysates of *Dnai1*^{-/-} mNPEC. To determine if it was due to the host species (i.e. mouse) used to develop the antibody, we also tried immunostaining using a rabbit anti-DNAI1 antibody that was used for immunoblots. Unfortunately, this antibody was not suitable to use for immunocytochemistry and was not successful in staining DNAI1 in our fixed mNPEC. Thus, we have not included these immunocytochemistry staining results in our manuscript.

4. Minor points - missing scale bar in Fig 1A and legends.

Thank you for pointing this out. We have included the missing scale bar in one of the merged images in Figure 1A. We have also the text “Scale grid for 3D projections, 40 µm.” or “Scale grid for 3D projections, 150 µm.” for Figure 1A and Figure 2C, respectively, on page 28 and 29.

Second decision letter

MS ID#: jcs.264068R1

MS Title: Lipid Nanoparticle-encapsulated *Dnai1* mRNA Rescues Ciliary Activity in Primary Ciliary Dyskinesia Mouse Cell Models

Authors: Amanda J. Smith; Patrick R. Sears; Mirko Hennig; Rumpa B. Bhattacharjee; Weining Yin; Hannah Golliher; Daniella Ishimaru; T. Noelle Lombana; David J. Lockhart; Brandon A. Wustman; Lawrence E. Ostrowski

Article Type: Research Article

Dear Amanda, Larry and team,

I am pleased to inform you that your manuscript has been accepted in principle for publication in our Special Issue on Cilia and Flagella for the Journal of Cell Science, pending standard publication integrity checks. The last thing is to look at the new Figure 2C ROI which needs a scale bar and accompanying grayscale HA channel zoom, as described below. It was accepted on 04 Sep 2025. Where referee reports on this version are available, they are appended below.